

# Generalized Komar charges and Smarr formulas for black holes and boson stars

**Romina Ballesteros[1,2]⋆ and Tomás Ortín[1]†**

**1** Instituto de Física Teórica UAM/CSIC, C/ Nicolás Cabrera 13–15,
C.U. Cantoblanco, E-28049 Madrid, Spain
**2** Pontificia Universidad Católica de Valparaíso, Instituto de Física,
Av. Brasil 2950, Valparaíso, Chile

⋆ romina.ballesteros@estudiante.uam.es , † tomas.ortin@csic.es

## Abstract

The standard Komar charge is a $(d-2)$-form that can be defined in spacetimes admitting a Killing vector and which is closed when the vacuum Einstein equations are satisfied. Its integral at spatial infinity (the Komar integral) gives the conserved charge associated to the Killing vector, and, due to its on-shell closedness, the same value (expressed in terms of other physical variables) is obtained integrating over the event horizon (if any). This equality is the basis of the Smarr formula. This charge can be generalized so that it still is closed on-shell in presence of matter and its integrals give generalizations of the Smarr formula. We show how the Komar charge and other closed $(d-2)$-form charges can be used to prove non-existence theorems for gravitational solitons and boson stars. In particular, we show how one can deal with generalized symmetric fields (invariant under a combination of isometries and other global symmetries) and how the generalized symmetric Ansatz permits to evade the non-existence theorems.

| | |
|---|---|
| Received | 2024-11-11 |
| Accepted | 2025-04-02 |
| Published | 2025-04-24 |

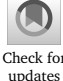

# 1   Introduction

Charge conservation is one of the more powerful ideas of Physics. Conserved charges can be used to label and characterize states or whole physical systems and their presence constrains their evolution and simplifies the study of their dynamics.

The conservation of many charges is associated, via Noether's theorem, to global symmetries. For each of those global symmetries there is a conserved current $j^\mu$ satisfying the continuity equation

$$\partial_\mu j^\mu = 0 \,. \tag{1}$$

The amount of conserved charge enclosed in a spatial volume $\Sigma^3$ is defined as

$$q \equiv \int_{\Sigma^{d-1}} d^3\Sigma_\mu j^\mu = \int_{\Sigma^{d-1}} d^{d-1}\Sigma n_\mu j^\mu \,, \tag{2}$$

where $n^\mu$ is the timelike unit vector normal to $\Sigma^{d-1}$. Then, the continuity equation implies that the variation in time of this charge equals the flux of this current across the boundary of the volume. In other words: the amount of charge in that volume at a given time equals the initial amount of charge plus or minus the amount of charge that entered or left the volume. The total charge of the Universe, then, must remain constant, because nothing can enter or leave it from outside (*i.e.* there are no sources nor sinks at infinity). We may characterize the Universe by the value of this charge.

It is this property (which is what we mean by *charge conservation*) that makes conserved charges real entities.

The continuity equation can be rewritten in a very convenient way in differential-form language, using the $(d-1)$-form $\mathbf{J} \equiv \star\left(j_\mu dx^\mu\right)$, which we will keep on calling *current*:

$$d\mathbf{J} = 0 \,. \tag{3}$$

The charge contained in the spatial volume $\Sigma^{d-1}$ is now

$$q = \int_{\Sigma^{d-1}} \mathbf{J} \,, \tag{4}$$

and charge conservation follows from Eq. (3) and Stokes theorem: if $\Sigma^d$ is the region of spacetime swept by the spatial volume $\Sigma^{d-1}$ in a given time and its boundary is

$$\partial \Sigma^d = \Sigma^{d-1}_{t_0} \cup \Sigma^{d-1}_{t_1} \cup \mathcal{B}_{t_0,t_1} \,, \tag{5}$$

integrating $d\mathbf{J}$ we find, taking into account the orientation of the different pieces

$$0 = \int_{\Sigma^d} d\mathbf{J} = \int_{\Sigma^{d-1}_{t_1}} \mathbf{J} - \int_{\Sigma^{d-1}_{t_0}} \mathbf{J} + \int_{\mathcal{B}_{t_0,t_1}} \mathbf{J} \,, \tag{6}$$

which leads to

$$q_{t_1} = q_{t_0} - \text{flux}^{t_1}_{t_0} \,, \tag{7}$$

where the flux

$$\text{flux}^{t_1}_{t_0} \equiv \int_{\mathcal{B}_{t_0,t_1}} \mathbf{J} \,, \tag{8}$$

is positive when the charge is leaving the volume.

Let us now consider the charges contained in the volume outside a black hole[1] at a given time $t_0$ and at a later time $t_1$.[2] The difference between these charges is (minus) the amount of charge that has crossed the horizon into the black hole. However, as a rule, black holes cannot carry charges whose conservation follows from global symmetries and, the net effect would be that the total amount of charge of the Universe would decrease. This kind of charges which are always computed via volume integrals cannot be used to characterize neither the black hole nor the whole black-hole spacetime. Furthermore, there are indications that Quantum Gravity may break all exact global symmetries. [1].

In some cases, the $(d-1)$-form currents are related on-shell or off-shell to $(d-2)$-form *charges*[3] by

$$\mathbf{J} = d\mathbf{Q} \,. \tag{9}$$

Sometimes, this relation only holds asymptotically. This is, actually, what happens with the gravitational field itself [2]. This relation presents us with new opportunities: one can replace volume integrals of $\mathbf{J}$ by surface integrals of $\mathbf{Q}$ over the boundaries or, in general, over closed $(d-2)$-surfaces $\Sigma^{d-2}$

$$q = \int_{\Sigma^{d-2}} \mathbf{Q} \,. \tag{10}$$

In black-hole spacetimes one can, then, do away with the problem of integrating inside the horizon and decree that the total charge of the spacetime is the integral of $\mathcal{Q}$ over the $(d-2)$-sphere at spatial infinity $\Sigma^{d-2} = S^{d-2}_{\infty}$. Gravitational charges are defined in this way and, as it is well known, it makes no sense to compute them integrating anywhere else because, due to the principle of Equivalence, it is not possible to define an energy/mass or momentum density function. Since there is no time evolution at spatial infinity, these charges characterize the whole spacetime.

In some cases the conserved charges obey "Gauss laws" because, at least in the spacetime region we are concerned with, $\mathbf{J} = 0$ and the $(d-2)$-form $\mathbf{Q}$ is closed, typically on-shell[4]

$$d\mathbf{Q} \doteq 0 \,. \tag{11}$$

---

[1]Extending the integral to the interior has many practical and conceptual problems.

[2]Here we must use Kruskal-Szekeres time, or other time coordinate which is well defined at the horizon, unlike Schwarzschild's.

[3]It is customary to use the same name for $q$ and for $\mathcal{Q}$. This does not lead to confusion, usually.

[4]We use $\doteq$ to denote identities which are only satisfied on-shell.

$(d-2)$-forms (and forms of other ranks) that satisfy this equation are often called conserved charges as well, although this equation is not equivalent to the continuity equation (3). Strictly speaking, this equation is just the differential expression of the Gauss law: due to the on-shell closedness of the $(d-2)$-form charge $\mathbf{Q}$, the value of the integral does not change under smooth deformations of the integration surfaces: if $\Sigma^{d-2\prime}$ has been obtained from $\Sigma^{d-2}$ by a smooth deformation (which includes the assumption that the deformation does not cross any point at which the classical equations of motion are not satisfied) and $\Sigma^{d-1\prime}$ is the cobordant volume

$$\partial\Sigma^{d-1\prime} = \Sigma^{d-2}\cup\Sigma^{d-2\prime}, \tag{12}$$

taking into account the orientations of $\Sigma^{d-2}$ and $\Sigma^{d-2\prime}$, Stokes theorem implies

$$\int_{\Sigma^{d-2}}\mathbf{Q} - \int_{\Sigma^{d-2\prime}}\mathbf{Q} = \int_{\Sigma^{d-1\prime}}d\mathbf{Q} \doteq 0, \tag{13}$$

which is the generalized form of the standard Gauss law of electromagnetism.

Gauss laws are typically satisfied by charges that source Abelian (uncharged) fields, such as the electric charge of standard electromagnetism. In those cases there is a very clear distinction between the sources of the fields and the fields themselves. In general, the charges of gravitational or non-Abelian Yang–Mills fields do not satisfy Gauss laws and only their values in the whole spacetime can be defined as we explained before. For instance, since the gravitational field carries energy, it is also the source of gravitational field and any smooth deformation of the integration surface will change the sources of gravitational field and consequently the value of the energy/mass enclosed by it. This implies that there can be no closed $(d-2)$-form charge describing a local energy density, as we mentioned before.

In spite of the above general discussion, there are special situations in which it is actually possible to define on-shell-closed $(d-2)$-form charges in theories of gravity [3]. The main example is that of spacetimes admitting a Killing vector $k$ in General Relativity without matter.[5] The associated on-shell-closed $(d-2)$-form is the *Komar charge* $\mathbf{K}[k]$ [5], which, in our notation and conventions,[6] takes the form

$$\mathbf{K}[k] = (-1)^{d-1}\frac{1}{16\pi G_N^{(d)}}\star(e^a\wedge e^b)P_{kab}, \tag{16}$$

where $P_{kab}$ is the *Lorentz momentum map* or *Killing bivector*

$$P_{kab} = \nabla_a k_b = \nabla_{[a}k_{b]}. \tag{17}$$

---

[5]More general on-shell and off-shell possibilities have been explored in Ref. [4].

[6]Our conventions are those of Ref. [45]. In particular, we use mostly minus signature and we will always describe the gravitational field through the Vielbein $e^a = e^a{}_\mu dx^\mu$, where Latin indices are tangent-space indices and Greek indices are coordinate-basis indices. Furthermore, our Levi–Civita spin connection $\omega^a{}_b = \omega_\mu{}^a{}_b dx^\mu$ and its curvature 2-form $R^a{}_b = \frac{1}{2}R_{\mu\nu}{}^a{}_b dx^\mu\wedge dx^\nu$ are defined through the relations

$$\mathcal{D}e^a \equiv de^a - \omega^a{}_b\wedge e^b = 0, \tag{14a}$$
$$R^a{}_b \equiv d\omega^a{}_b - \omega^a{}_c\wedge\omega^c{}_b, \tag{14b}$$

where $\mathcal{D}$ is the exterior Lorentz-covariant derivative.

Equivalent ways of writing this $(d-2)$-form (ignoring the overall $(16\pi G_N^{(d)})$ factor) using the levi-Civita affine connection and its associated covariant derivative $\nabla$ are

$$\begin{aligned}(-1)^{d-1}\star(e^a\wedge e^b)P_{kab} &= (-1)^{d-1}\frac{\varepsilon_{\mu_1\cdots\mu_{d-2}\alpha\beta}}{(d-2)!\sqrt{|g|}}\nabla^\alpha k^\beta dx^{\mu_1}\wedge\cdots\wedge dx^{\mu_{d-2}}\\ &= d^{d-2}\Sigma_{\alpha\beta}\nabla^\alpha k^\beta\\ &= -d^{d-2}\Sigma n_{\alpha\beta}\nabla^\alpha k^\beta,\end{aligned} \tag{15}$$

where $n_{\alpha\beta}$ is the binormal to the horizon, normalized as $n_{\mu\nu}n^{\mu\nu} = -2$.

More details on our notation and conventions can be found in Ref. [45].

In vacuum,

$$d\mathbf{K}[k] \doteq 0\,. \tag{18}$$

The integral of this $(d-2)$-form charge over a $(d-2)$-sphere at spatial infinity $S_\infty^{d-2}$ (often called *Komar integral*) gives the conserved gravitational charge associated with the Killing vector $k$, up to normalization. For instance, in stationary spacetimes, the Komar integral associated to the timelike Killing vector $\partial_t$ gives the mass of the spacetime

$$\int_{S_\infty^{d-2}} \mathbf{K}[\partial_t] = \frac{d-3}{d-2}M\,, \tag{19}$$

while the Komar integral associated to the Killing vector that generates rotations around some symmetry axis, $\partial_\varphi$, gives the component of the angular momentum in that direction

$$\int_{S_\infty^{d-2}} \mathbf{K}[\partial_\varphi] = J\,. \tag{20}$$

Since the Komar charge is closed, the same results can be obtained by integrating over any other $(d-2)$ surface that can be smoothly taken to infinity. If one chooses spheres, the integrands are independent of the radial coordinate. In particular, in stationary black-hole spacetimes one can integrate over any section of the event horizon obtaining exactly the same results. Different choices of section correspond to different choices of hypersurface $\Sigma^{d-1}$.

What makes this seemingly trivial result really interesting is that the integral over the horizon is naturally expressed in terms of the physical variables associated to the horizon, such as the Hawking temperature $T$ and Bekenstein-Hawking entropy $S$, giving rise to an identity (the *Smarr formula* [6]) that relates, in a highly non-trivial way these two sets of physical variables [7–15].[7]

Let us show how the Smarr formula can be obtained in this way.

If the horizon $\mathcal{H}$ is non-degenerate, it is most convenient to choose the bifurcation surface $\mathcal{BH}$ because it leads to great simplifications. In particular, if $k = \partial_t - \Omega_H \partial_\varphi$ is the Killing vector that becomes null on it, $k^2 \overset{\mathcal{H}}{=} 0$, one can show that, on the bifurcation surface $\mathcal{BH}$

$$P_{kab} \overset{\mathcal{BH}}{=} \kappa n_{ab}\,, \tag{21}$$

where $\kappa$ is the surface gravity. Then, using Eq. (15) and the zeroth law of black-hole mechanics ($d\kappa \overset{\mathcal{H}}{=} 0$) [7] we get, integrating over the bifurcation surface

$$\int_{\mathcal{BH}} \mathbf{K}[k] = -\frac{\kappa}{16\pi G_N^{(d)}} \int_{\mathcal{BH}} d^{(d-2)}\Sigma\, n_{ab} n^{ab} = \frac{\kappa A_H}{8\pi G_N^{(d)}} = TS\,. \tag{22}$$

On the other hand, using Eqs. (19) and (20) the integral at spatial infinity gives

$$\int_{S_\infty^{d-2}} d\mathbf{K}[k] = \int_{S_\infty^{d-2}} \mathbf{K}[\partial_t] - \Omega_H \int_{S_\infty^{d-2}} \mathbf{K}[\partial_\varphi] = \frac{d-3}{d-2}M - \Omega_H J\,, \tag{23}$$

where we have also used the fact that $\Omega_H$ is constant over the horizon, which can be understood as a generalization of the zeroth law.

The on-shell-closedness of the Komar charge tells us that the results of these two integrals are equal

$$M = \frac{d-2}{d-3}\left(TS + \Omega_H J\right)\,, \tag{24}$$

---

[7]Other methods have been used to derive Smarr formulas. For instance, in Ref. [16] the reduction of the action to that of a $\sigma$-model was exploited to this end. See also Ref. [17].

which is the Smarr formula for stationary, asymptotically flat black holes in vacuum.

The Komar charge Eq. (16) is no longer closed on-shell when gravity is coupled to most forms of matter and the above algorithm cannot be used directly to derive Smarr formulas. One can follow the procedure outlined in Ref. [18] (see, for instance, Ref. [19]) involving volume integrals but these do not give directly expressions involving the total, conserved ADM charges. The derivation of the Smarr formulas is, therefore, not as clean as the one pioneered by Bardeen, Carter and Hawking [7,8] that we have just explained here.

The Komar charge of General Relativity in absence of matter is just (minus)[8] the Noether $(d-2)$-form charge associated to the invariance of the theory under diffeomorphisms (also known as *Noether–Wald charge*) $\mathbf{Q}[\xi]$ evaluated on a Killing vector $k$ which leaves invariant the only field of this theory, the metric, that is

$$\delta_k g_{\mu\nu} = -\pounds_k g_{\mu\nu} = 0\,, \tag{25}$$

where $\pounds_k$ is the Lie derivative.

This suggests that (minus) the Noether–Wald charge of the matter-coupled General Relativity theory, $-\mathbf{Q}[\xi]$, evaluated over vector fields $k$ generating symmetries of all the fields of the theory that we denote generically by $\varphi$,

$$\delta_k \varphi = 0\,, \tag{26}$$

may provide the on-shell-closed generalization of the Komar charge Eq. (16) that we need. Observe that the above condition implies Eq. (25) and $k$ of a Killing vector of the spacetime metric.

However, as observed in Ref. [20] (see also Ref. [21]), in most diffeomorphism-invariant theories[9]

$$d\mathbf{Q}[k] \doteq \iota_k \mathbf{L}\,, \tag{27}$$

where $\mathbf{L}$ is the $d$-form Lagrangian and $\iota_k \mathbf{L}$ is its interior product with the Killing vector field $k$. Integrating both sides of this equation over a hypersurface $\Sigma^{d-1}$ with boundaries at infinity and at the bifurcation surface

$$\partial\Sigma^{d-1} = S_\infty^{d-2} \cup \mathcal{BH}\,, \tag{28}$$

and applying Stokes theorem

$$\int_{\Sigma^{d-1}} d\mathbf{Q}[k] = \int_{S_\infty^{d-2}} \mathbf{Q}[k] - \int_{\mathcal{BH}} \mathbf{Q}[k] \doteq \int_{\Sigma^{d-1}} \iota_k \mathbf{L}\,. \tag{29}$$

This relation can be used to obtain generalized Smarr formulas [20],[10] but the volume integral term obscures their interpretation as relations between physical quantities defined at the horizon and spatial infinity.

In Ref. [15] it was argued that, as long as the diffeomorphism generated by $k$ leaves invariant all the fields of the theory[11] for a given solution, and, hence, leaves invariant the Lagrangian evaluated on-shell, $\iota_k \mathbf{L}$ is an exact $(d-1)$-form. Indeed,

$$0 \doteq \delta_k \mathbf{L} = -\pounds_k \mathbf{L} = -d\iota_k \mathbf{L}\,, \tag{30}$$

---

[8]In our conventions.

[9]In theories with Chern–Simons terms, invariant under gauge transformations up to total derivatives, there could be additional terms in the right-hand side of this expression. This is due to the fact, to be explained shortly, that diffeomorphisms induce gauge transformations.

[10]See also the more recent Ref. [21], based on the results of Ref. [22].

[11]When the theory contains fields with gauge freedoms, the transformations generated by $k$, $\delta_k$, acting on them have to be defined carefully so that the statements $\delta_k \varphi = 0$ are gauge invariant [23–25]. This point will be explained in more detail shortly.

implies the local existence of a $(d-2)$-form $\omega_k$ such that

$$\iota_k \mathbf{L} \doteq d\omega_k \,, \tag{31}$$

and we can define the generalized Komar $(d-2)$-form charge

$$\mathbf{K}[k] \equiv -(\mathbf{Q}[k] - \omega_k) \,, \tag{32}$$

which is closed on-shell by construction.

In many [26–30] but not all [31] theories it is possible to give an explicit expression of $\omega_k$ which just needs to be evaluated on a given solution. In all the cases we have studied so far the algorithm gives a Smarr formula with terms which are products of a thermodynamic charge $q$ times its conjugate chemical potential $\Phi$. These objects appear in the first law in terms of the form $\Phi\delta q$ and sometimes they have to be understood in the context of extended thermodynamics.

The use of the generalized Komar charge is not necessarily restricted to the derivation of Smarr formulas for black holes.[12] Here we want to use it to study stationary, globally regular, horizonless, topologically trivial, asymptotically flat solutions of General Relativity coupled to bosonic fields that we will loosely call "boson stars", although this name is used in a slightly more restricted sense in the literature (see, for instance, Refs. [19,33] and references therein). These solutions can be physically understood as self-gravitating solitons of some bosonic fields whose density is not high enough to cause gravitational collapse.

As a simple example of what we intend to do, let us consider possible boson star solutions of General Relativity in vacuum. The boundary of any Cauchy hypersurface of these spaces is the $(d-2)$-sphere at infinity. Integrating $d\mathbf{K}[\partial_t] \doteq 0$ over the Cauchy hypersurface and using Stokes theorem and Eq. (19) leads to $M = 0$ so that, according to the positive mass theorem [34–36], the solution must be Minkowski spacetime. In other words: there are no boson stars in vacuum.

This example shows how the generalized Komar charge can be used to constrain the possible solutions of a given theory.[13] It also tells us that, in order to make non-trivial solutions possible we have to relax one or several of these assumptions: stationarity, regularity, asymptotic flatness, topological triviality, absence of event horizons and Einstein equations in vacuum.

We have already seen that a non-degenerate event horizon works as a second, inner boundary[14] on which the Komar integral of the Killing vector that generates it gives $TS$, so that $M - \Omega_H J = 2TS \geq 0$, which can be satisfied for $M \neq 0$, even if the horizon is degenerate and $T = 0$.

In absence of horizons, for obvious reasons, we do not want to give up on the regularity of the solutions and, since we are interested in stationary, asymptotically-flat solutions of trivial

---

[12]Scalar charges satisfying Gauss laws have also been used to prove no-hair theorems in Ref. [29] (see Section 2) and to prove that certain initial data sets do not correspond to constant-time slices of stationary black-hole solutions in Ref. [32]. We will use these and other charges in our analysis.

[13]The study of the (non-)existence of non-topological solitons has a long history. The first result of this kind is Derrick's theorem Ref. [37] which proves the non-existence of particle-like solutions of non-linear generalizations of the wave equation. The possibility of the existence of those solutions if they were allowed to have a periodic time dependence was also suggested in that reference. Solutions of the Einstein-Klein-Gordon theory of this kind were first found by Kaup in Ref. [39]. The characterization of non-gravitating spherically-symmetric solitons (called there *Q-balls*) by the charge associated to a non-gauged global symmetry (a U(1) symmetry, typically) and their existence were studied by Coleman in Ref. [38]. It is this symmetry that is used to make a generalized symmetric Ansatz, as we will explain. Stars and black holes with solitonic scalar fields having a U(1) global symmetry were constructed by Friedberg, Lee and Pang in Ref. [40].

[14]This is not completely true: we can always choose a hypersurface $\Sigma^{d-1}$ that goes inside the horizon. However, in general, inside the horizon we are going to find singularities or non-trivial topology, including other asymptotically-flat regions. If we do not want to deal with these, we must consider another boundary and a section of the horizon (its bifurcation surface, if any) is the most convenient place for it due to its special properties.

topology, we can only play with two assumptions: the coupling to different kinds of matter (so we do not deal with the vacuum Einstein equations) and with the implementation of the stationarity assumption on the matter fields, which can modify the Komar charge. Most often, we deal with axisymmetric spacetimes. The condition of axisymmetry can also be implemented in different ways.

This requires an explanation.

An asymptotically-flat spacetime is called stationary when it admits a Killing vector field which is asymptotically timelike and axisymmetric when it admits a Killing field vector with periodic orbits and fixed points (the axis). Let us call $k$ any of these vector fields. The conditions of invariance are expressed by Eq. (25) above. If this metric is part of a solution that includes matter fields it is commonly assumed that they should also be exactly invariant under the diffeomorphism generated by $k$ as well. This is usually expressed as

$$\delta_k \varphi = -\pounds_k \varphi = 0\,. \tag{33}$$

Most theories are invariant under local and global symmetries which act on the same fields and we must take this fact into account in the above expression. Let us start with gauge symmetries.

## 1.1 Spacetime symmetries of fields with gauge freedoms

In general, gauge symmetries do not commute with the standard Lie derivative and the above expression is, at the very least, ambiguous, because it is not gauge-invariant. Using it can lead to wrong results [25] and, as repeatedly argued in Refs. [23–25] it must be replaced by a *gauge-covariant Lie derivative* $\mathbb{L}_k$ that annihilates the fields in a gauge-invariant way for the Killing vector $k$. This covariant Lie derivative is always a combination of the standard one $\pounds_k \varphi$ and a field- and Killing-vector-dependent gauge transformation[15] of the field $\delta_{\Lambda(k,\varphi)}\varphi$. Thus, we are led to use the transformation

$$\delta_k \varphi = -\mathbb{L}_k \varphi = -\left(\pounds_k - \delta_{\Lambda(k,\varphi)}\right)\varphi = 0\,. \tag{34}$$

The gauge transformation $\delta_{\Lambda(k,\varphi)}$ can be seen as "induced" by the diffeomorphism generated by the Killing vector.

An alternative way of looking at this transformation is to consider that the best we can do with fields with gauge freedoms is to ask for invariance under diffeomorphisms up to gauge transformations

$$\pounds_k \varphi = \delta_{\Lambda(k,\varphi)}\varphi\,. \tag{35}$$

While the first point of view is actually closer to the rigorous description of the action of diffeomorphisms in fiber bundles,[16] the second is going to help us to understand better the implementation of the stationarity condition on fields transforming under some global symmetries of the theory.

## 1.2 Spacetime symmetries of fields with global freedoms

Let us denote the action of the independent global symmetries by $\delta_I \varphi$ where $I,J,K,\dots$ label them. Indeed, looking at the above transformation, it is very natural to generalize the standard *symmetric* Ansatz Eq. (33) to[17]

$$\pounds_k \varphi = \vartheta_k{}^I \delta_I \varphi\,, \tag{36}$$

---

[15]The field-dependent parameters of these gauge transformations must be the same for all the fields transforming under the same symmetry, though.

[16]See, *e.g.* [41, 42].

[17]For simplicity we consider only gauge-invariant fields.

where the $\vartheta_k{}^I$ are constants that depend on the Killing vector $k$[18] and which must be compatible with the global structure (the periodicity, for instance, see footnote 22 in page 11) of the symmetries involved.

For scalar fields, for instance, the above *generalized symmetric* Ansatz allows dependence on the coordinate adapted to the isometry generated by $k$, in contrast to what happens to the metric. It can be shown, however, that the generalized symmetric Ansatz leads to exactly symmetric energy-momentum tensors $\pounds_k T_{\mu\nu} = 0$ [43] so that it is perfectly compatible with the isometry. More generally, if $T$ is any tensor invariant under the global symmetry,

$$\pounds_k T = \vartheta_k{}^I \delta_I T = 0. \tag{37}$$

There is an important point concerning the generalized symmetry Ansatz Eq. (36) that we would like to clarify here. Most of the General Relativity literature considers that the generalized symmetric Ansatz for the scalar fields prevents the spacetime isometries from becoming true symmetries of the complete field configuration. A common way of expressing this idea is by saying that the scalar fields "do not inherit" the symmetry of the spacetime metric.[19] However, it follows from our previous discussion of the action of isometries on fields with gauge symmetries that they do not "inherit" spacetime symmetries in the usual, restricted, sense in which spacetime symmetries act on all fields through standard Lie derivatives. Still, there is a perfectly well defined (and gauge-invariant) sense in which the diffeomorphism generated by the vector $k$ leaves the field invariant, expressed through the combination of the standard Lie derivative and the "compensating" gauge transformation into the gauge-covariant Lie derivative. Gauge fields may be time-dependent in a certain gauge and, still, be invariant under the gauge-covariant Lie derivative with respect to $\partial_t$ so that this isometry is a symmetry of all the fields of the theory and not just of the metric. The solution is stationary even if the gauge fields have dependence on $t$. Similar remarks apply to axisymmetry and the dependence on the angular coordinate $\varphi$

The same reasoning can be used in the case of global symmetries: the fields are now invariant under the transformation

$$\delta_k \varphi \equiv -\left(\pounds_k - \vartheta_k{}^I \delta_I\right)\varphi = 0, \tag{38}$$

so that $\delta_k$ is a symmetry of all the fields of the theory, although it does not simply act as the Lie derivative with respect to $k$ on all the fields. For $k = \partial_t$, each possible choice of the constants $\vartheta_k{}^I$ provides a different implementation of the stationarity condition but for any choice, there is a well-defined sense in which we can still say that the solution is stationary, even if the matter fields have a dependence on $t$.

All the boson star solutions found so far are based on non-trivial choices of $\vartheta_k{}^I$s for $\partial_t$ and $\partial_\varphi$ in stationary and axisymmetric spacetimes. One of our goals is to understand this fact using the generalized Komar charge which must be modified by those choices.

In this paper we are going to consider different kinds of matter coupled to gravity and different implementations of the stationarity and axisymmetry Ansatzs on them, constructing the generalized Komar charge and other charges satisfying Gauss laws in each case and finding the implications for the existence of horizonless, globally regular, topologically trivial, asymptotically flat solutions in $d = 4$ dimensions. It is organized as follows: In Section 2 we consider the simplest kind of matter, namely a single, real, massless scalar field. Next, in Section 3 we consider the Einstein–Maxwell theory. In Section 4 we add a scalar potential to the theory of Section 2, breaking its shift symmetry. In Section 5 we consider theories with an arbitrary

---

[18]Here we are considering just one isometry and the index $k$ in $\vartheta_k{}^I$ is redundant but it will be relevant when we consider more general situations with several isometries in Appendix A.

[19]See, for instance Ref. [44] and references therein.

number of scalars and Abelian vector fields which have the generic form of a 4-dimensional, ungauged supergravity theory, combining and generalizing the results of the previous sections. Section 6 contains a discussion of our results. Appendix A contains a general discussion of the generalized symmetric Ansatz using only scalar fields as an example: consistency conditions (very similar to the famous quadratic constraint of the embedding tensor formalism) and the definition of scalar charges. Appendix B reviews the proof of the generalized zeroth law for electrostatic and magnetostatic potentials that we use in the main text. Finally, in Appendix C we show that solutions satisfying the generalized symmetric Ansatz with respect to electric-magnetic duality rotations (a symmetry of the equations of motion but not of the action) can be found, even though they turn out not to be very interesting in this simple case.

## 2 The Einstein–scalar theory

The simplest kind of matter that can be coupled to Einstein's gravitational field which we will always describe through the Vierbein $e^a = e^a{}_\mu dx^\mu$, is a real, massless scalar $\phi$. The action for this Einstein–scalar (ES) theory is

$$S[e,\phi] = \frac{1}{16\pi G_N^{(4)}} \int \left[ -\star(e^a \wedge e^b) \wedge R_{ab} + \tfrac{1}{2} d\phi \wedge \star d\phi \right] \equiv \int \mathbf{L}. \qquad (39)$$

If the scalar field satisfies the standard symmetric Ansatz Eq. (33) for the timelike Killing vector $k = \partial_t$ it is not difficult to see that the generalized Komar charge of this theory is equal to the standard one given in Eq. (16) (see, for instance, Ref. [31]). This implies that there are no boson star solutions[20] in this theory with the standard implementation of the stationarity condition. On the other hand, according to the Smarr formula Eq. (24), the presence of a non-degenerate horizon allows for black holes of mass $M = 2ST + 2\Omega_H J$.

Before we study whether the non-trivial implementations of the stationarity and axisymmetry conditions through the generalized symmetric Ansatz can modify this conclusion, it is convenient to define a charge for the scalar $\phi$ that satisfies a Gauss law in a stationary spacetime [29, 47]. Since the theory is invariant under global shifts of the scalar field, there is an on-shell-conserved Noether current whose Hodge-dual 3-form we denote by $J_\phi$ and which is given by

$$J_\phi \equiv \frac{1}{16\pi G_N^{(4)}} \star d\phi. \qquad (40)$$

The on-shell conservation is here equivalent to the on-shell-closedness of $J_\phi$:

$$dJ_\phi = -\mathbf{E}_\phi \doteq 0, \qquad (41)$$

where $\mathbf{E}_\phi$ is the equation of motion of $\phi$.

The charge associated to the above 3-form, given by its integral over a 3-dimensional spacelike hypersurface,[21] though, does not satisfy a Gauss law. Furthermore, in stationary spacetimes in which the scalar is time-independent it vanishes identically [29] because the timelike component of the current does. A different charge is needed to characterize the scalar field.

If all the fields are exactly invariant under the diffeomorphism generated by $k$, so is $J_\phi$ and, using Cartan's magic formula and Eq. (41)

$$0 = \delta_k J_\phi = -\pounds_k J_\phi = -(\iota_k d + d\iota_k)J_\phi \doteq d\iota_k J_\phi, \qquad (42)$$

---

[20]As discussed in the Introduction, we are using here the term "boson star" in a rather loose way. This is probably more evident in this case and more conventional or proper way of expressing this result would be the non-existence of non-topological solitons [46].

[21]This is equivalent to the volume integral of the timelike component of the standard current.

which tells us that the 2-form $\mathbf{Q}_\phi[k] = \iota_k J_\phi$ is closed on-shell and the charge $\Sigma_\phi$ defined by its integral over closed spacelike surfaces satisfies a Gauss law [29, 47]. In standard spherical coordinates $\Sigma$ appears as the coefficient of the $1/r$ term in the asymptotic expansion of $\phi$.

The same arguments used for the Komar charge show that boson stars would be characterized by $\Sigma_\phi = 0$ which is consistent with their non-existence.

In presence of a non-degenerate horizon $\mathcal{H}$, using the Killing vector $k$ normal to it ($k^2 \overset{\mathcal{H}}{=} 0$) and integrating $d\mathbf{Q}_\phi[k] = 0$ over a hypersurface satisfying Eq. (28), taking into account that, by definition, $k \overset{\mathcal{BH}}{=} 0$ so that $\mathbf{Q}_\phi[k] \overset{\mathcal{BH}}{=} 0$, we find again $\Sigma_\phi = 0$. This result can be interpreted as a no-hair theorem relating the existence of a non-degenerate horizon to the absence of scalar charge of the kind we have defined above [29].

## 2.1 The Einstein–scalar theory and the generalized symmetric Ansatz

Now we want to see how the previous results may be modified if we implement the stationarity condition ($k = \partial_t$) in the form[22]

$$\delta_k \phi = -(\pounds_k \phi - \vartheta_k) = 0 \,, \tag{43}$$

allowing for linear dependence of the scalar on the time coordinate:

$$\phi = \vartheta_k t + f(x^1, x^2, x^3) \,. \tag{44}$$

Notice that at any given time $t = t_0$ it is possible to eliminate the $\vartheta_k t_0$ using the shift symmetry. The term $\vartheta_k t$ is only relevant globally.

In this setting, the Noether current associated to the invariance of the theory under constant shifts of the scalar, $J_\phi$, defined in Eq. (40) does not vanish any longer.

The Noether–Wald charge of the theory, $\mathbf{Q}[\xi]$, is associated to the invariance of the action under diffeomorphisms only and, therefore, it is not modified by the generalized symmetric Ansatz. It has the same expression as in the vacuum theory, namely

$$\mathbf{Q}[\xi] = \frac{1}{16\pi G_N^{(4)}} \star (e^a \wedge e^b) P_{\xi\,ab} \,. \tag{45}$$

By construction,
$$d\mathbf{Q}[\xi] = \mathbf{\Theta}(e^a, \phi, \delta_\xi e^a, \delta_\xi \phi) + \mathbf{E}_a \xi^a + \iota_\xi \mathbf{L} \,, \tag{46}$$

where the presymplectic potential is given by

$$\mathbf{\Theta}(e^a, \phi, \delta_\xi e^a, \delta_\xi \phi) = \star(e^a \wedge e^b) \wedge \left(\mathcal{D}P_\xi{}^{ab} + \iota_\xi R^{ab}\right) - \star d\phi \, \iota_\xi d\phi \,. \tag{47}$$

When $\xi = k$ the term in parenthesis vanishes identically.[23] With the standard symmetric Ansatz, the second term vanishes as well and, on-shell, we are left with $d\mathbf{Q}[k] \doteq \iota_k \mathbf{L}$, but in

---

[22]Observe that this Ansatz is consistent because both the scalar field $\phi$ and the time coordinate $t$ take values in $\mathbb{R}$. In axisymmetric spacetimes, whose metric is invariant under constant shifts of the angular coordinate $\varphi \sim \varphi + 2\pi$, the Ansatz $\partial_\varphi \phi = \vartheta_\varphi$ leads to a multivalued scalar field unless it is assumed that $\phi$ must also be identified with $\phi + 2\pi\vartheta_\varphi$. This is not possible in this theory, but in theories with more scalar fields that can be understood as coordinates in a "target space", some of them may also be understood as angular, periodically identified, coordinates and the Ansatz would be consistent. This is the case of the phase of the complex Klein–Gordon scalar, for instance [48, 49]. We will consider this possibility in Section 5.3.

[23]The equation

$$\mathcal{D}P_k{}^{ab} + \iota_k R^{ab} = 0 \,, \tag{48}$$

defines the Lorentz momentum map $P_k{}^{ab}$ and is solved when $P_k{}^{ab} = \nabla_a k_b$, the Killing bivector. Actually, replacing the Lorentz momentum map by the Killing bivector this equation becomes the integrability condition of the Killing vector equation. On the other hand, the left-hand side of the equation is just $-\delta_k \omega^{ab}$ [23].

this theory $\mathbf{L} \doteq 0$ and the Komar charge coincides with (minus) the Noether–Wald charge and with that of the vacuum theory.

In the generalized symmetric case Eq. (43) the Lagrangian and the first term in the presymplectic potential still vanish on-shell, but the second term does not and

$$\mathbf{\Theta}(e^a, \phi, \delta_k e^a, \delta_k \phi) = -\vartheta_k \star d\phi = -\vartheta_k J_\phi \,, \tag{49}$$

so that

$$d\mathbf{Q}[k] \doteq -\vartheta_k J_\phi \,. \tag{50}$$

Since $J_\phi$ is closed on-shell, locally there must exist a 2-form charge $\mathbf{Q}_\phi$ such that

$$J_\phi \doteq d\mathbf{Q}_\phi \,, \tag{51}$$

and we can define the on-shell-closed generalized Komar charge

$$\mathbf{K}[k] \equiv -\left(\mathbf{Q}[k] + \vartheta_k \mathbf{Q}_\phi\right) \,, \tag{52}$$

which differs from the one obtained with the standard implementation of the stationarity condition.

In this case there is no general expression for $\mathbf{Q}_\phi$: it depends on the explicit form of the solution. Its actual form will not be important in what follows, though.

Let us consider a boson star solution and let us integrate $d\mathbf{K}[k] \doteq 0$ over a spacelike hypersurface bounded by a 2-sphere of radius $r$. Applying Stokes theorem we find that

$$\int_{S_r^2} \mathbf{Q}[k] \doteq -\vartheta_k \int_{\Sigma_r^2} \mathbf{Q}_\phi \,. \tag{53}$$

The integral in the left-hand side converges to $M/2$ as $r$ approaches infinity. The integral in the right-hand side gives the scalar charge contained in the 2-sphere and it is easy to see that this charge diverges when $r$ approaches infinity. This indicates that boson stars of this kind do not exist in this theory. The situation is not improved by admitting the existence of an event horizon or rotation since the integral that gives the charge keeps diverging at infinity. Thus, stationary, asymptotically flat black-hole solutions of this theory do not exist, either. Coupling the scalar field to itself or to some other source can change these conclusions, as we are going to see in Section 4.

For black-hole spacetimes there is another way to arrive at this result which does not rely on these charges and which will play an important role later. If $k$ is the Killing vector that becomes null on the horizon $k^2 \overset{\mathcal{H}}{=} 0$ one can show that

$$k^\mu k^\nu R_{\mu\nu} \overset{\mathcal{H}}{=} 0 \,, \tag{54}$$

which implies, upon use of the Einstein equations and $k^2 \overset{\mathcal{H}}{=} 0$

$$k^\mu k^\nu T_{\mu\nu} \overset{\mathcal{H}}{=} 0 \,, \tag{55}$$

leading, in this particular theory, to the conclusion

$$\iota_k d\phi \overset{\mathcal{H}}{=} 0 \,, \tag{56}$$

so that $\vartheta_k = 0$. If the black hole is rotating $k = \partial_t - \Omega \partial_\varphi$ and, since in this case $\partial_\varphi \phi = 0$ (see footnote 22 in page 11), we arrive at the same conclusion as before. Notice, though, that in the cases in which it is consistent to impose the generalized symmetric Ansatz

$$\partial_t \phi = \vartheta_t \equiv \omega \,, \tag{57a}$$

$$\partial_\varphi \phi = \vartheta_\varphi \equiv m \,, \tag{57b}$$

with non-vanishing $\omega$ and $m$, Eq. (56) demands

$$\frac{\omega}{m} = \Omega_H\,,\tag{58}$$

known as "synchronization condition" [19], on account of the relation $\vartheta_k = \omega - \Omega_H m$.

## 3 The Einstein–Maxwell theory

The Maxwell (or electromagnetic) field provides $A = A_\mu dx^\mu$ another simple kind of matter that can be coupled to Einstein's gravitational field. It is worth stressing that, despite the notation, geometrically $A$ is not a 1-form, but a gauge field, a connection in a U(1) fiber bundle, with gauge transformations

$$\delta_\chi A = d\chi\,,\tag{59}$$

where $\chi$ is any real function. Its gauge-invariant 2-form field strength, locally given by

$$F \equiv dA,\tag{60}$$

is a 2-form, though.

The action of the 4-dimensional Einstein–Maxwell (EM) theory is

$$S[e,A] = \frac{1}{16\pi G_{\mathrm{N}}^{(4)}} \int \left[ -\star(e^a \wedge e^b) \wedge R_{ab} + \tfrac{1}{2}F \wedge \star F \right] \equiv \int \mathbf{L}\,.\tag{61}$$

Notice that this theory does not contain any fields charged with respect to the Maxwell field that can source it. The action is not invariant under any global transformations of the Maxwell field only, but the equations of motion supplemented with the Bianchi identity are invariant under an SO(2) group of electric-magnetic duality rotations and we are going to show in Section 3.1 that they may be consistently used in the generalized symmetric Ansatz. This theory is too simple for this Ansatz to give rise to boson star solutions but we can consider it as a "proof of concept" that opens the door to the use in more complicated theories such as those we are going to in Section 5.

As we have explained in the introduction,[24] the stationarity condition of the Maxwell field has to be implemented in a gauge-invariant form, combining the standard Lie derivative with a gauge transformation with parameter $\chi_k = \iota_k A - P_k$ where the *momentum map* $P_k$ satisfies the gauge-invariant *momentum map equation*

$$\iota_k F + dP_k = 0\,,\tag{62}$$

whose integrability condition is the symmetry condition

$$\delta_k F = -\pounds_k F = -\iota_k F = 0\,.\tag{63}$$

The gauge-invariant form of stationarity condition of the Maxwell field is the *momentum map equation*

$$\delta_k A = -\left(\pounds_k - \delta_{\chi_k}\right)A = -\left(\iota_k F + dP_k\right) = 0\,.\tag{64}$$

Observe that, if $k$ is timelike, $\iota_k F$ is the electric field for an observer related to the time defined by $k$ and, therefore, $P_k$ is the associated electrostatic potential $\Phi$ defined, as usual, up to an additive constant. In asymptotically-flat spacetimes $\Phi$ takes a constant value at infinity $\Phi_\infty$ that is purely conventional.

---

[24]See Ref. [23] for details on this particular case.

In this theory it is possible to find a generic form of $\omega_k$ and the generalized Komar charge of this theory is [23, 26, 28]

$$\mathbf{K}[k] = -\frac{1}{16\pi G_N^{(4)}} \star (e^a \wedge e^b) P_{kab} + \frac{1}{32\pi G_N^{(4)}} \left[ P_k \star F - \tilde{P}_k F \right], \tag{65}$$

where we have introduced the dual momentum map $\tilde{P}_k$, defined by the *dual momentum map equation*

$$\iota_k \star F + d\tilde{P}_k = 0, \tag{66}$$

whose local existence is guaranteed on-shell by the stationarity condition. It can also be understood as the magnetostatic potential $\tilde{\Phi}$ and, in asymptotically-flat spacetimes it takes an arbitrary constant value at infinity $\tilde{\Phi}_\infty$.

Apart from the generalized Komar charge Eq. (65) the EM theory has another two interesting 2-form charges:

$$\mathbf{Q} = \frac{1}{16\pi G_N^{(4)}} \star F, \tag{67}$$

which is closed on-shell and whose integral over a closed 2-surface gives the electric charge $q$ enclosed by it, and

$$\mathbf{P} = \frac{1}{16\pi G_N^{(4)}} F, \tag{68}$$

which is closed off-shell and whose integral over a closed 2-surface gives the magnetic charge $p$ enclosed by it. Notice that, Stokes theorem implies that $p$ would vanish identically if $F$ was $dA$ globally.

Since the theory does not contain sources of the Maxwell field, we expect any non-trivial stationary, asymptotically flat[25] electromagnetic fields to be sourced by singularities or sustained by non-trivial topology. Indeed, the usual arguments applied to $\mathbf{Q}$ and $\mathbf{P}$ in boson stars lead to $q = p = 0$.

Let us now turn our attention to the generalized Komar charge which can be rewritten in the form

$$\mathbf{K}[k] = -\frac{1}{16\pi G_N^{(4)}} \star (e^a \wedge e^b) P_{kab} + \tfrac{1}{2} \left[ \Phi \mathbf{Q} - \tilde{\Phi} \mathbf{P} \right]. \tag{69}$$

Its integral over the 2-sphere at spatial infinity gives

$$M + \Phi_\infty q - \tilde{\Phi}_\infty p = 0, \tag{70}$$

which reduces to $M = 0$ once the vanishing of the total electric and magnetic charges has been taken into account. Again, solutions of the kind we are looking for do not exist in the EM theory. Observe that the vanishing of $q$ and $p$ is crucial for the consistency of the result because the values of $\Phi_\infty$ and $\tilde{\Phi}_\infty$ can be changed arbitrarily keeping the metric (and $M$) unchanged.

It is interesting to see how the presence of a non-degenerate horizon modifies this conclusion.

First of all, we must replace $\partial_t$ by the Killing vector $k = \partial_t - \Omega_H \partial_\varphi$ that becomes null on the horizon. The momentum maps $P_k$ and $\tilde{P}_k$ cannot be interpreted as purely electrostatic and magnetostatic potentials, although we will keep the notation $\Phi, \tilde{\Phi}$. Moreover, they do not tend to just a constant value at infinity but their $r \to \infty$ limits may contain, for instance, terms proportional to $\Omega_H \cos\theta$, in addition to the arbitrary constants $\Phi_\infty, \tilde{\Phi}_\infty$. Fortunately, it can be

---

[25]Asymptotic flatness excludes homogeneous gravitational/electromagnetic waves which can be stationary, regular, horizonless and topologically trivial.

proved that those terms do not contribute to the integrals of $\Phi \mathbf{Q}$ and $\tilde{\Phi} \mathbf{P}$ at infinity [50] and the integral at infinity of the generalized Komar charge Eq. (69) gives

$$\tfrac{1}{2}\left(M + \Phi_\infty q - \tilde{\Phi}_\infty p\right) - \Omega_H J, \tag{71}$$

while the integral on the bifurcation surface gives

$$TS + \tfrac{1}{2}\left(\Phi_H q - \tilde{\Phi}_H p\right), \tag{72}$$

where $\Phi_H$ and $\tilde{\Phi}_H$ are the values of the electrostatic and magnetostatic in $\mathcal{BH}$ which are constant by virtue of the *restricted generalized zeroth law* [23] which we review in Appendix B.

Combining these results we obtain the Smarr formula for stationary, asymptotically flat black holes of the EM theory

$$M = 2ST + 2\Omega_H J + (\Phi_H - \Phi_\infty)q - \left(\tilde{\Phi}_H - \tilde{\Phi}_\infty\right)p. \tag{73}$$

Notice that the non-vanishing electric and magnetic charges multiply unambiguous differences of potentials.

## 3.1 Generalized symmetric Ansatz

As we have mentioned before, the (left-hand side of the) Einstein equations $\mathbf{E}_a$, Maxwell equations $\mathbf{E}$ and Bianchi identities $\mathbf{B}$, given by[26]

$$\mathbf{E}_a = \iota_a \star (e^b \wedge e^c) \wedge R_{bc} + \tfrac{1}{2}\left(\iota_a F \wedge \tilde{F} - F \wedge \iota_a \tilde{F}\right), \tag{74a}$$
$$\mathbf{E} = -d\tilde{F}, \tag{74b}$$
$$\mathbf{B} = -dF, \tag{74c}$$

where we have defined the dual 2-form field strength

$$\tilde{F} \equiv \star F. \tag{75}$$

These transformations act on $F$ ad $\tilde{F}$ as

$$\begin{pmatrix} F \\ \tilde{F} \end{pmatrix}' = \begin{pmatrix} \cos\alpha & \sin\alpha \\ -\sin\alpha & \cos\alpha \end{pmatrix} \begin{pmatrix} F \\ \tilde{F} \end{pmatrix}, \qquad \delta_\alpha \begin{pmatrix} F \\ \tilde{F} \end{pmatrix} = \begin{pmatrix} 0 & \alpha \\ -\alpha & 0 \end{pmatrix} \begin{pmatrix} F \\ \tilde{F} \end{pmatrix}. \tag{76}$$

Thus, we may implement the stationarity Ansatz on these fields in a non-trivial way, using this global symmetry, as follows:

$$\pounds_k \begin{pmatrix} F \\ \tilde{F} \end{pmatrix} = \begin{pmatrix} 0 & \alpha \\ -\alpha & 0 \end{pmatrix} \begin{pmatrix} F \\ \tilde{F} \end{pmatrix}. \tag{77}$$

Notice that this Ansatz guarantees the invariance of the Maxwell energy-momentum tensor $T_a$ given by

$$T_a \equiv \tfrac{1}{2}\left(\iota_a F \wedge \tilde{F} - F \wedge \iota_a \tilde{F}\right), \tag{78}$$

under the diffeomorphism generated by the Killing vector $k$, $\pounds_k T_a = 0$.

It is convenient to introduce a dual (*magnetic*) gauge field $\tilde{A}$ such that, locally,[27]

$$\tilde{F} \equiv d\tilde{A}. \tag{79}$$

---

[26]We ignore the overall factors of $(16\pi G_N^{(4)})^{-1}$ in order to simplify the expressions.

[27]We are not going to address any global issues related to the existence of $A$ or $\tilde{A}$ in this section.

The pair $A, \tilde{A}$ transform under electric-magnetic duality as the pair $F, \tilde{F}$, namely,

$$\delta_\alpha \begin{pmatrix} A \\ \tilde{A} \end{pmatrix} = \begin{pmatrix} 0 & \alpha \\ -\alpha & 0 \end{pmatrix} \begin{pmatrix} A \\ \tilde{A} \end{pmatrix}. \tag{80}$$

It is worth remarking that, although this is not a symmetry of the action, there is an on-shell conserved current associated to it:[28]

$$J_{em} \equiv \tfrac{1}{2} \left( A \wedge F + \tilde{A} \wedge \tilde{F} \right), \qquad dJ_{em} \doteq 0, \tag{82}$$

which, furthermore, is invariant under electric-magnetic duality transformations.[29]

Then, the generalized symmetric Ansatz Eq. (77) is locally equivalent to the conditions

$$d \begin{pmatrix} \iota_k F - \alpha \tilde{A} \\ \iota_k \tilde{F} + \alpha A \end{pmatrix} = 0, \tag{84}$$

which imply the local existence of the electric and magnetic momentum maps $P_k, \tilde{P}_k$ satisfying the *generalized momentum maps equations*

$$\begin{pmatrix} \iota_k F \\ \iota_k \tilde{F} \end{pmatrix} = -D \begin{pmatrix} P_k \\ \tilde{P}_k \end{pmatrix}, \tag{85}$$

where we have defined the covariant derivatives

$$D \begin{pmatrix} P_k \\ \tilde{P}_k \end{pmatrix} \equiv d \begin{pmatrix} P_k \\ \tilde{P}_k \end{pmatrix} - \begin{pmatrix} 0 & \alpha \\ -\alpha & 0 \end{pmatrix} \begin{pmatrix} A \\ \tilde{A} \end{pmatrix}, \tag{86}$$

which are invariant under the gauge transformations

$$\delta_\chi \begin{pmatrix} A \\ \tilde{A} \end{pmatrix} = d \begin{pmatrix} \chi \\ \tilde{\chi} \end{pmatrix}, \qquad \delta_\chi \begin{pmatrix} P_k \\ \tilde{P}_k \end{pmatrix} = \begin{pmatrix} 0 & \alpha \\ -\alpha & 0 \end{pmatrix} \begin{pmatrix} \chi \\ \tilde{\chi} \end{pmatrix}. \tag{87}$$

The momentum maps transform as Stückelberg fields and we may eliminate them using the appropriate gauge transformation, if necessary. Thus, it is clear that they cannot be identified with electrostatic or magnetostatic potentials as in the symmetric case.

To check the consistency of this construction, it is interesting to review it in terms of the gauge fields. Taking into account their gauge freedoms, the generalized symmetric Ansatz reads

$$\pounds_k \begin{pmatrix} A \\ \tilde{A} \end{pmatrix} = d \begin{pmatrix} \chi_k \\ \tilde{\chi}_k \end{pmatrix} + \begin{pmatrix} 0 & \alpha \\ -\alpha & 0 \end{pmatrix} \begin{pmatrix} A \\ \tilde{A} \end{pmatrix}, \tag{88}$$

and we can immediately see that it is satisfied by the choice of parameters of the compensating gauge transformations

$$\begin{pmatrix} \chi_k \\ \tilde{\chi}_k \end{pmatrix} = \begin{pmatrix} \iota_k A - P_k \\ \iota_k \tilde{A} - \tilde{P}_k \end{pmatrix}. \tag{89}$$

---

[28] The conservation is due to the well-known property

$$\star B \wedge \star A = -A \wedge B, \tag{81}$$

for any pair of 2-forms $A, B$ in 4 dimensions. This property is, precisely the responsible for the non-invariance of the Maxwell action under electric-magnetic duality transformations.

[29] There is another electric-magnetic duality-invariant current

$$J_{em-2} \equiv \tfrac{1}{2} \left( A \wedge \tilde{F} - \tilde{A} \wedge F \right) = d \left( \tfrac{1}{2} \tilde{A} \wedge A \right), \tag{83}$$

but it does not seem to play any role in what follows.

In black-hole spacetimes, if $k = \partial_t - \Omega_H \partial_\varphi$ is the Killing vector that characterizes the event horizon as a Killing horizon, in the definition of the generalized symmetric Ansatz we have to use the same global symmetry for $\partial_t$ (with parameter $\omega$) and $\partial_\varphi$ (with parameter $m$) and the covariant derivatives of the momentum maps are now

$$D\begin{pmatrix} P_k \\ \tilde{P}_k \end{pmatrix} \equiv d\begin{pmatrix} P_k \\ \tilde{P}_k \end{pmatrix} - (\omega - \Omega_H m)\begin{pmatrix} 0 & 1 \\ -1 & 0 \end{pmatrix}\begin{pmatrix} A \\ \tilde{A} \end{pmatrix}. \tag{90}$$

On the horizon, both $\iota_k F$ and $\iota_k \tilde{F}$ vanish identically. In the symmetric case the generalized zeroth law can be derived from this fact (see Appendix B). In this case, we find that

$$D\begin{pmatrix} P_k \\ \tilde{P}_k \end{pmatrix} \overset{\mathcal{H}}{=} 0. \tag{91}$$

If the synchronization condition Eq. (58) is not satisfied $\omega - \Omega_H m \neq 0$, these equations imply that both connections $A$ and $\tilde{A}$ are pure gauge on the horizon and, thus, the field strength and its dual vanish identically there

$$\begin{pmatrix} F \\ \tilde{F} \end{pmatrix} \overset{\mathcal{H}}{=} 0. \tag{92}$$

The generalized symmetric Ansatz Eq. (77) does not modify the definitions of electric and magnetic charges satisfying Gauss laws $\mathbf{Q}, \mathbf{P}$ Eqs. (67) and (68). Since we can compute the electric and magnetic charges integrating these field strengths over any section of the horizon obtaining the same result (because it is the same result one obtains integrating over the 2-sphere at spatial infinity), we conclude that, in this case, the generalized symmetric Ansatz leads to vanishing electric and magnetic charges $q = p = 0$ in black-hole spacetimes. In boson-star spacetimes they vanish for the same reasons as in the previous cases.

If the synchronization condition Eq. (58) is satisfied, then the momentum maps admit the standard interpretation of electrostatic and magnetostatic potentials and satisfy the generalized zeroth law and the electric and magnetic charges need not vanish.

The generalized Komar charge is modified. The Noether–Wald charge $\mathbf{Q}[k]$ only depends on diffeomorphisms and the induced gauge transformations and it is not modified and it is still given by [23]

$$\mathbf{Q}[\xi] = \star(e^a \wedge e^b)P_{\xi\,ab} - P_\xi \tilde{F}. \tag{93}$$

On-shell, it satisfies

$$d\mathbf{Q}[\xi] \doteq \mathbf{\Theta}(e, A, \delta_\xi e, \delta_\xi A) + \iota_\xi \mathbf{L}, \tag{94}$$

where the presymplectic potential and the on-shell Lagrangian are given by

$$\mathbf{\Theta}(e, A, \delta_\xi e, \delta_\xi A) = -\star(e^a \wedge e^b) \wedge \delta_\xi \omega_{ab} - \tilde{F} \wedge \left(\iota_\xi F + dP_\xi\right), \tag{95a}$$

$$\mathbf{L} \doteq \tfrac{1}{2}F \wedge \tilde{F}. \tag{95b}$$

Now, when $\xi = k$, the first term in $\mathbf{\Theta}$ vanishes but the second does not, according to the generalized momentum map equations (85), which we also have to apply to the calculation of $\iota_k \mathbf{L}$. Substituting the result in $d\mathbf{Q}[k]$, we get

$$d\left[-\star(e^a \wedge e^b)P_{k\,ab} + \tfrac{1}{2}\left[P_k \tilde{F} - \tilde{P}_k F\right]\right] \doteq -\omega J_{em}, \tag{96}$$

where $J_{em}$ has been defined in Eq. (82). Since $J_{em}$ is closed on-shell, there must be a 2-form $\mathcal{J}_{em}$, whose form depends on the particular solution on which we evaluate it, such that

$$d\mathcal{J}_{em} \doteq J_{em}, \tag{97}$$

and we arrive at the following generalized Komar charge

$$\mathbf{K}[k] = -\star(e^a \wedge e^b)P_{kab} + \tfrac{1}{2}\left[P_k\tilde{F} - \tilde{P}_kF\right] + \omega\mathcal{J}_{em}. \tag{98}$$

Since the electric and magnetic charges must vanish for any hypothetical boson star, it seems unlikely that the integral of $\mathcal{J}_{em}$ at infinity could help us to avoid the conclusion $M = 0$. In Appendix C we have studied, as a proof of concept, a simple example of time-dependent solutions of the Maxwell equations in a (non-back-reacted) stationary spacetime (Minkowski spacetime), showing their existence. They turn out to be superpositions of electromagnetic waves with no electric nor magnetic charges.

In black hole spacetimes the last term must be replaced by $(\omega - \Omega_H m)\mathcal{J}_{em}$ and there are two possible cases: when the synchronization condition is not satisfied we must use the vanishing of $F$ and $\tilde{F}$ on the bifurcation surface because the momentum maps do not satisfy a restricted generalized zeroth law and, again, the vanishing of the electric and magnetic charges makes it very unlikely that the current $\mathcal{J}_{em}$ gives any finite contribution at infinity. When the synchronization condition is satisfied, the last term vanishes identically and the momentum maps are constant over the horizon and we recover exactly the same Smarr formula as in the symmetric case. It remains to be seen if there are any black hole solutions satisfying this Ansatz but, in principle, the Smarr formula does not exclude this possibility.

## 4 The Einstein–scalar theory with a scalar potential

The thermodynamics of the black holes of this theory was recently studied by us in Ref. [31]. Thus, we shall be brief. The action is the one considered in Eq. (39) plus a scalar potential term:

$$S[e,\phi] = \frac{1}{16\pi G_N^{(4)}} \int \left\{ -\star(e^a \wedge e^b) \wedge R_{ab} + \tfrac{1}{2}d\phi \wedge \star d\phi + \star V(\phi) \right\} \equiv \int \mathbf{L}. \tag{99}$$

The scalar potential $V(\phi)$ is assumed not to be constant, so it cannot be interpreted as a cosmological constant. Therefore, there is no shift symmetry, no associated on-shell conserved Noether current and no possible generalized symmetric Ansatz in this theory. In spite of this, an on-shell-closed 2-form charge can be defined in stationary solutions using the following observation [29,47]: if all the fields of the solution we are considering are invariant under the diffeomorphism generated by $k$, we have, on the one hand

$$-\iota_k d \star d\phi = (d\iota_k - \pounds_k) \star d\phi = d\iota_k \star d\phi, \tag{100}$$

and on the other hand

$$0 = \pounds_k \star V' = d\iota_k \star V', \qquad V' \equiv \frac{\partial V}{\partial \phi}, \tag{101}$$

which implies the local existence of a 2-form $\mathcal{W}_k$ such that

$$\iota_k \star V' \doteq d\mathcal{W}_k. \tag{102}$$

In this case it is not possible to find a generic expression for $\mathcal{W}_k$. Its form will depend on the particular solution on which $\star V'$ is evaluated but we can always add a closed 2-form to it so that, for asymptotically flat solutions $\mathcal{W}_k(\infty) = 0$.

Then, if we take the interior product of $k$ with the scalar equation of motion and use the above results

$$\begin{aligned}\iota_k\mathbf{E}_\phi &= \frac{1}{16\pi G_N^{(4)}}\iota_k\left[-d\star d\phi + \star V'\right] \\ &= d\left\{\frac{1}{16\pi G_N^{(4)}}\left[\iota_k \star d\phi + \mathcal{W}_k\right]\right\},\end{aligned} \tag{103}$$

and we can define the on-shell closed 2-form charge[30]

$$\mathbf{Q}_\phi[k] \equiv -\frac{1}{4\pi G_N^{(4)}}\left[\iota_k \star d\phi + \mathcal{W}_k\right],\tag{104}$$

whose integral over closed 2-dimensional surfaces gives, by definition, the scalar charge $\Sigma$ enclosed in it. At infinity, because of the boundary condition $\mathcal{W}_k(\infty) = 0$, the scalar charge depends only on the first term in the above equation. Furthermore, the usual argument leads to $\Sigma = 0$. In presence of a non-degenerate horizon we find that $\Sigma$ is given by the integral of $\mathcal{W}_k$ on the bifurcation sphere. Thus, $\Sigma$ would be secondary hair, dependent on the dimensionful constants defining $V$.

The generalized Komar charge is in this theory [29]

$$\mathbf{K}[k] = -\frac{1}{16\pi G_N^{(4)}}\left[\star(e^a \wedge e^b)P_{kab} - \mathcal{V}_k\right],\tag{105}$$

where, in the same vein as $\mathcal{W}_k$, the 2-form $\mathcal{V}_k$ is defined to satisfy

$$d\mathcal{V}_k \doteq -\iota_k \star V.\tag{106}$$

Its value at spatial infinity can be set to zero for asymptotically-flat solutions, $\mathcal{V}_k(\infty) = 0$, upon the addition of a closed 2-form.

With the chosen boundary conditions, the integral of $\mathbf{K}[k]$ at spatial infinity gives, yet again, $M = 0$. When the potential is not definite-positive, this may not be enough to ensure that the only possible solution is Minkowski spacetime because the positive mass theorem would not be valid.

Allowing for a non-degenerate horizon and choosing $k$ accordingly, we get the Smarr formula [29]

$$M = 2ST + 2\Omega_H J + 2\alpha\Phi_\alpha,\quad \text{where}\quad \Phi_\alpha \equiv -\frac{1}{16\pi G_N^{(4)}}\int_{\Sigma^3}\iota_k \star \frac{\partial V}{\partial\alpha},\tag{107}$$

where $\alpha$ is a dimensionful constant in $V$ which plays the role of a new thermodynamic variable while $\Phi_\alpha$ plays the role of its conjugate potential [15, 27, 51]. If the scalar potential depends on more dimensionful constants, $\alpha^i$, the last term is replace by a sum over similar terms.

## 5 The Einstein–Maxwell–scalar theories

In this section we are going to consider a generic supergravity-inspired theory that includes the three cases we have studied so far as particular examples. This theory contains, apart from the gravitational field $e^a$, $n_v$ Abelian gauge fields $A^\Lambda$, $\Lambda = 1, \ldots, n_V$. with field strengths $F^\Lambda \equiv dA^\Lambda$ and $n_S$ scalar fields $\phi^x$, $x = 1, \ldots, n_s$ that parametrize a non-linear $\sigma$-model with positive-definite target-space metric $g_{xy}(\phi)$ coupling to themselves via a scalar potential $V(\phi)$ and to the Abelian gauge fields via the symmetric, scalar-dependent, matrices $I_{\Lambda\Sigma}(\phi)$ (which

---

[30]The normalization is purely conventional.

is conventionally assumed to be negative-definite) and $R_{\Lambda\Sigma}(\phi)$. The action takes the form[31]

$$S = \frac{1}{16\pi G_N^{(4)}} \int \left[ -\star(e^a \wedge e^b) \wedge R_{ab} + \tfrac{1}{2} g_{xy} d\phi^x \wedge \star d\phi^y \right.$$
$$\left. -\tfrac{1}{2} I_{\Lambda\Sigma} F^\Lambda \wedge \star F^\Sigma - \tfrac{1}{2} R_{\Lambda\Sigma} F^\Lambda \wedge F^\Sigma + \star V \right]. \tag{108}$$

The equations of motion $\mathbf{E}_a, \mathbf{E}_x, \mathbf{E}_\Lambda$ and the presymplectic potential $\Theta(\varphi, \delta\varphi)$ (here $\varphi$ stands for all the fields of the theory) are defined by the general variation of the fields in the action

$$\delta S = \int \left\{ \mathbf{E}_a \wedge \delta e^a + \mathbf{E}_x \delta\phi^x + \mathbf{E}_\Lambda \delta A^\Lambda + d\Theta(\varphi, \delta\varphi) \right\}. \tag{109}$$

Ignoring the overall factors of $(16\pi G_N^{(4)})^{-1}$, they are given by

$$\mathbf{E}_a = \iota_a \star (e^b \wedge e^c) \wedge R_{bc} + \tfrac{1}{2} g_{xy} \left( \iota_a d\phi^x \star d\phi^y + d\phi^x \wedge \iota_a \star d\phi^y \right)$$
$$-\tfrac{1}{2} I_{\Lambda\Sigma} \left( \iota_a F^\Lambda \wedge \star F^\Sigma - F^\Lambda \wedge \iota_a \star F^\Sigma \right) - \iota_a \star V, \tag{110a}$$

$$\mathbf{E}_x = -g_{xy} \left\{ d \star d\phi^y + \Gamma_{zw}{}^y d\phi^z \wedge \star d\phi^w \right\}$$
$$-\tfrac{1}{2} \partial_x I_{\Lambda\Sigma} F^\Lambda \wedge \star F^\Sigma - \tfrac{1}{2} \partial_x R_{\Lambda\Sigma} F^\Lambda \wedge F^\Sigma + \star \partial_x V, \tag{110b}$$

$$\mathbf{E}_\Lambda = dF_\Lambda, \tag{110c}$$

$$\Theta(\varphi, \delta\varphi) = -\star(e^a \wedge e^b) \wedge \delta\omega_{ab} + g_{xy} \star d\phi^x \delta\phi^y - F_\Lambda \wedge \delta A^\Lambda, \tag{110d}$$

where we have defined the dual 2-form field strength

$$F_\Lambda \equiv I_{\Lambda\Sigma} \star F^\Sigma + R_{\Lambda\Sigma} F^\Sigma. \tag{111}$$

## 5.1 Global symmetries

In supergravity theories the symmetries of the $\sigma$-model, generated by the Killing vectors $K_I{}^x(\phi)$ of $g_{xy}(\phi)$

$$\delta_I g_{xy} = -\pounds_{K_I} g_{xy} = 0, \tag{112}$$

and the symmetries of the Abelian gauge fields are related[32] thanks to the equivariance property

$$\delta_I R_{\Lambda\Sigma} = T_{I\,\Lambda\Sigma} + T_{I\Lambda}{}^\Omega R_{\Omega\Sigma} - R_{\Lambda\Omega} T_I{}^\Omega{}_\Sigma - R_{\Lambda\Gamma} T_I{}^{\Gamma\Omega} R_{\Omega\Sigma} + I_{\Lambda\Gamma} T_I{}^{\Gamma\Omega} I_{\Omega\Sigma}, \tag{113a}$$

$$\delta_I I_{\Lambda\Sigma} = T_{I\Lambda}{}^\Omega I_{\Omega\Sigma} - I_{\Lambda\Omega} T_I{}^\Omega{}_\Sigma - 2 R_{(\Lambda|\Gamma} T_I{}^{\Gamma\Omega} I_{\Omega|\Sigma)}. \tag{113b}$$

We will assume that this is the case here. Otherwise we would restrict ourselves to the subgroup of isometries of the $\sigma$-model metric for which the above equivariance conditions hold.

---

[31]In supergravity theories, the numbers $n_V$ and $n_s$, the $\sigma$-model metric, the scalar matrices and the scalar potential are constrained by supersymmetry. In supergravity, the scalar potential would be associated to the gauging of R-symmetry. The gauging of other symmetries would give rise to couplings not concluded in the action Eq. (108). In particular, the gauging of symmetries which act as isometries of the $\sigma$-model metric leads to the replacement of the derivatives of the scalar fields by covariant derivatives. In some cases the gauge symmetries can be used to completely gauge away some of the scalars, whose kinetic terms become mass terms for the vector fields. The so-called Einstein–Proca–Higgs model considered in Refs. [19, 52] can be obtained in this way starting from a model of the form Eq. (108). We will study this more general class of models elsewhere [53].

[32]See, e.g. Section 1 of Ref. [29] for a detailed description of the properties of the scalar matrices that are required for this relation to work.

Thus, in absence of the scalar potential, which may break some of those symmetries, the equations of motion are invariant under the following transformations of the scalars and gauge fields

$$\delta_I \phi^x = K_I{}^x(\phi),$$ (114a)

$$\delta_I F^M = T_I{}^M{}_N F^N,$$ (114b)

where we have defined the symplectic vectors of field strengths

$$\left(F^M\right) \equiv \begin{pmatrix} F^\Lambda \\ F_\Lambda \end{pmatrix},$$ (115)

and where the matrices $T_I$ in Eqs. (113a) are blocks of the matrices $T_I{}^M{}_N$ in Eq. (114b), *i.e.*

$$\left(T_I{}^M{}_N\right) = \begin{pmatrix} T^\Lambda{}_\Sigma & T^{\Lambda\Sigma} \\ T_{\Lambda\Sigma} & T_\Lambda{}^\Sigma \end{pmatrix}.$$ (116)

The Killing vectors $K_I$ and the $2n_V \times 2n_V$ matrices $T_I$ satisfy the Lie algebra

$$[K_I, K_J] = -f_{IJ}{}^K K_K,$$ (117a)

$$[T_I, T_J] = +f_{IJ}{}^K T_K.$$ (117b)

Furthermore, the matrices $T_I$ belong to the Lie algebra of the symplectic group $SL(2n_V, \mathbb{R})$ in the fundamental (vector) representation [54], which implies for the block matrices

$$\begin{aligned} T_{A\Lambda\Sigma} &= T_{A\Sigma\Lambda}, \\ T_A{}^\Lambda{}_\Sigma &= -T_{A\Sigma}{}^\Lambda, \\ T_A{}^{\Lambda\Sigma} &= T_A{}^{\Sigma\Lambda}. \end{aligned}$$ (118)

Some of the transformations of the field strengths are electric-magnetic duality transformations transforming $F^\Lambda$ into $F_\Lambda$

$$\delta_I F^\Lambda = T_I{}^\Lambda{}_\Sigma F^\Sigma + T_I{}^{\Lambda\Sigma} F_\Sigma.$$ (119)

Those transformations are symmetries of the equations of motion but they do not leave the action invariant. As we have seen in Section 3.1 they can also be used to formulate a generalized symmetric Ansatz.

In presence of an scalar potential, we are going to assume that a subset of these symmetries whose generators we will label with $A, B, C, \ldots$ ($K_A, T_A$) generate a Lie subalgebra with structure constants $f_{AB}{}^C$ are preserved. In particular, we assume that

$$\pounds_{K_A} V = K_A{}^x \partial_x V = 0.$$ (120)

We are going to denote with indices $U, V, W, \ldots$ the Killing vectors that do not leave invariant the scalar potential

$$\pounds_{K_U} V = K_U{}^x \partial_x V \neq 0.$$ (121)

## 5.2 Charges and Smarr formula for the symmetric Ansatz

Contracting the scalar equations of motion $\mathbf{E}_x$ with the Killing vectors $K_I{}^x$ we get [55]:

$$K_I{}^x \mathbf{E}_x \equiv dJ_I + \star K_I{}^x \partial_x V + \tfrac{1}{2}\Omega_{MP} T_A{}^P{}_N A^M \wedge \mathbf{E}^N,$$ (122a)

$$J_I = -K_{Ix} \star d\phi^x - \tfrac{1}{2}\Omega_{MP} T_I{}^P{}_N A^M \wedge F^N,$$ (122b)

where we have used Eqs. (112), (113a) and (120) and where

$$(\Omega_{MN}) = \begin{pmatrix} 0 & \mathbb{1}_{n_V \times n_V} \\ -\mathbb{1}_{n_V \times n_V} & 0 \end{pmatrix}. \tag{123}$$

In absence of scalar potential, the 3-forms $J_I$ would be the on-shell-closed *Noether–Gaillard–Zumino (NGZ)* currents of the theory [54]. The scalar potential breaks some of the global symmetries and we end up with the on-shell-closed NGZ currents $J_A$ and the currents $J_U$ which are not closed on-shell:

$$dJ_A \doteq 0, \tag{124a}$$

$$dJ_U \doteq - \star K_U{}^x \partial_x V. \tag{124b}$$

In stationary solutions in which all the fields satisfy the symmetric Ansatz (invariance without compensating global transformations[33]), the usual symmetry arguments lead to two kinds of on-shell-closed 2-form charges [29]

$$\mathbf{Q}_A[k] \equiv \frac{1}{4\pi G_N^{(4)}} \left\{ \iota_k K_{Ax} \star d\phi^x + \Omega_{MP} T_A{}^P{}_N P_k{}^M F^N \right\}, \tag{125a}$$

$$\mathbf{Q}_U[k] \doteq \frac{1}{4\pi G_N^{(4)}} \left\{ \iota_k K_{Ux} \star d\phi^x + \Omega_{MP} T_U{}^P{}_N P_k{}^M F^N + \mathcal{W}_{U,k} \right\}, \tag{125b}$$

where the 2-forms $\mathcal{W}_{U,k}$ are defined by

$$d\mathcal{W}_{U,k} \equiv \iota_k \star K_U{}^x \partial_x V, \tag{126}$$

and where we have defined the symplectic vector of momentum maps

$$\left( dP_k{}^M \right) \equiv \left( \iota_k F^M \right). \tag{127}$$

As in the case considered in Section 4, we cannot give a generic expression for the 2-forms $\mathcal{W}_{U,k}$ because it will depend on the particular solution considered.

The generalized Komar charge of this theory (symmetric Ansatz) can be easily found combining the results and strategies of the previous sections with those of Refs. [26, 28][34] and it is given by

$$\mathbf{K}[k] = -\frac{1}{16\pi G_N{}^{(4)}} \left\{ \star(e^a \wedge e^b) P_{kab} + \tfrac{1}{2} \left[ P_k{}^\Lambda F_\Lambda - P_{k\Lambda} F^\Lambda \right] - \mathcal{V}_k \right\}, \tag{128}$$

where $\mathcal{V}_k$ has been defined in Eq. (106).

Finally, the electric $q_\Lambda$ and magnetic $p^\Lambda$ charges, combined in a symplectic vector of charges $q^M$ can be defined as the integrals of the symplectic vector of 2-form charges

$$\mathbf{Q}^M \equiv \frac{1}{16\pi G_N^{(4)}} F^M, \qquad q^M \equiv \int_{S_\infty^2} \mathbf{Q}^M. \tag{129}$$

As usual, with the trivial implementation of the stationarity condition, electric, magnetic and scalar charges must vanish $q^M = \Sigma_A = \Sigma_U = 0$. Taking this into account, the integral of the exterior derivative of the Komar charge over a Cauchy surface of a boson star gives, yet again, $M = 0$.

Allowing for a non-degenerate horizon and choosing $k$ accordingly, we get a generalization of the Smarr formulas obtained before, that combines them:

$$M = 2ST + 2\Omega_H J + (\Phi_H^\Lambda - \Phi_\infty^\Lambda) q_\Lambda - (\Phi_{\Lambda H} - \Phi_{\Lambda \infty}) p^\Lambda + 2\alpha \Phi_\alpha, \tag{130}$$

where $\alpha$ and $\Phi_\alpha$ have the same meaning as in Eq. (107).

---

[33]As we have stressed, for the gauge fields, we always need to introduce a "compensating" gauge transformation.

[34]The derivation is reviewed to account for the generalized symmetric Ansatz in Section 5.3, anyway.

### 5.3 Generalized symmetric Ansatz

The theories we are considering can have a large number of global symmetries, with different kinds of orbits. This means that, in stationary, axisymmetric spacetimes, it may be possible to define a generalized symmetric Ansatz involving the Killing vector $\partial_\varphi$ as well as $\partial_t$:[35]

$$\partial_t \phi^x = \vartheta_t{}^A K_A{}^x \equiv \omega K^x \,, \tag{131a}$$

$$\partial_\varphi \phi^x = \vartheta_\varphi{}^A K_A{}^x(\phi) \equiv m L^x \,, \tag{131b}$$

for two constants $\omega, m$ and two Killing vectors of the target space metric $K, L$ that also leave invariant the scalar potential $V$. In the particular case of the massive, complex, Klein–Gordon field there is only one isometry that leaves invariant the scalar potential and, therefore, $K = L$ was the only possibility considered in Ref. [48, 49].

Since the global symmetries of these theories also act on the gauge field strengths and on their duals according to Eq. (114b) we must also include them in the Ansatz:

$$\pounds_t F^M = \vartheta_t{}^A T_A{}^M{}_N F^N \equiv \omega T^M{}_N F^N \,, \tag{132a}$$

$$\pounds_\varphi F^M = \vartheta_\varphi{}^A T_A{}^M{}_N F^N \equiv m S^M{}_N F^N \,. \tag{132b}$$

Since $\partial_t$ and $\partial_\varphi$ have vanishing spacetime Lie brackets, the matrices $T$ and $S$ must commute and $K$ and $L$ must have vanishing target space Lie brackets as well. Thus, we can use target space coordinates (scalar fields) adapted to them. If we call them, respectively, $\phi^1$ and $\phi^2$, the Ansatz reads

$$\partial_t \phi^x = \omega \delta^x{}_1 \,, \tag{133a}$$

$$\partial_\varphi \phi^x = m \delta^x{}_2 \tag{133b}$$

$$\Rightarrow \quad \phi^x = \delta^x{}_1 \omega t + \delta^x{}_2 m \varphi + f^x(x^1, x^2) \,, \tag{133c}$$

and, if $K = L$, then $\phi^2 = \phi^1$, $T = S$ and

$$\partial_t \phi^x = \omega \delta^x{}_1 \,, \tag{134a}$$

$$\partial_\varphi \phi^x = m \delta^x{}_1 \tag{134b}$$

$$\Rightarrow \quad \phi^x = \delta^x{}_1 (\omega t + m \varphi) + f^x(x^1, x^2) \,. \tag{134c}$$

As we have mentioned before, it is usually stated that the solutions satisfying this Ansatz are neither stationary nor antisymmetric and that their only spacetime symmetry, present when $K = L$ is the one generated by the Killing vector

$$k \equiv \partial_t - \frac{\omega}{m} \partial_\varphi \,. \tag{135}$$

As we have explained at length in the introduction, the above Ansatz follows from a modified ("twisted") definition of what being stationary and axisymmetric means. In other words, it is a different implementation of stationarity and axisymmetry and we are going to see that one can define on-shell closed Komar charges for each Killing vector $\partial_t$ and $\partial_\varphi$, independently.

---

[35]See footnote 22 in page 11. Notice that this is the most general thing we can do: as proven in Appendix A.1, if we only use the Killing vector $\partial_\varphi$ the spacetime cannot be spherically symmetric nor it can admit any other Killing vector whose Lie brackets with $\partial_\varphi$ do not vanish. In black-hole spacetimes this is usually associated to rotation.

The combination $k$ in Eq. (135) is clearly special, though, and it can be shown[36] that, the event horizon of stationary, axisymmetric black holes satisfying this Ansatz (if any) is the Killing horizon of $k$, which implies the "synchronization condition" Eq. (58) Ref. [19] which can only be satisfied in the case $K = L$, $\phi^1 = \phi^2$.

As for the gauge fields, following the same steps as in Section 3.1, defining the gauge fields $A^M$, $F^M = dA^M$, we find that the generalized Ansatz implies

$$\iota_{t,\varphi} F^M = -DP_{t,\varphi}{}^M , \tag{139}$$

where we have defined the gauge-covariant derivatives

$$DP_t{}^M \equiv dP_t{}^M - \omega T^M{}_N A^N , \qquad DP_\varphi{}^M \equiv dP_\varphi{}^M - mS^M{}_N A^N , \tag{140}$$

invariant under

$$\delta_\chi A^M = d\chi^M , \qquad \delta_\chi P_t{}^M = \omega T^M{}_N \chi^N , \qquad \delta_\chi P_\varphi{}^M = mS^M{}_N \chi . \tag{141}$$

Notice that, in black-hole spacetimes, for $k = \partial_t - \Omega_H \partial_\varphi$ and using Eq. (138) we find that

$$\omega T^M{}_N - \Omega_H mS^M{}_N = 0 , \tag{142}$$

which implies $T = S$ and the synchronization condition. Then, in the Einstein–Maxwell case the momentum maps satisfy the standard momentum map equation (127) which leads to the generalized zeroth law for the electrostatic and magnetostatic potentials $\Phi^M = P_k{}^M$.

Let us now consider the definitions of the different charges in this setup.

Electric and magnetic charges are still defined by Eq. (129) and satisfy Gauss laws. Thus, they are doomed to vanish identically in boson stars, but not necessarily in black-hole spacetimes.

Let us now consider the scalar charges. The standard symmetry arguments leading to the expression Eq. (125a) are valid and lead to the same expressions except for the charges associated to the generators $K, L$ and $T, S$ involved in the generalized symmetry Ansatz. Then, except in these two cases, those charges must vanish in boson stars and are subject to no-hair theorems in asymptotically-flat black-hole spacetimes [29]. The two exceptions fall in the general case considered in Appendix A.2 and, since the global symmetries they are associated with commute, they have the same form as in the standard case and must also vanish for boson stars and are subject to no-hair theorems.

The charges $\mathbf{Q}_{Uk}$ do not need to vanish, though, and should be treated as in Section 4 [31].

Let us now consider the Komar charge. The Noether–Wald charge can be found as in the symmetric case and takes the form [29]

$$\mathbf{Q}[\xi] = \star(e^a \wedge e^b) P_{\xi\,ab} + P_\xi{}^\Lambda F_\Lambda . \tag{143}$$

---

[36]The proof is based on Eq. (55), which can be seen to apply separately to the energy-momentum tensor of the scalar fields and of the gauge fields: since $\iota_k \iota_k F^\Lambda = 0$, we find that

$$\iota_k F^\Lambda \overset{\mathcal{H}}{=} f^\Lambda \hat{k} + \hat{v}^\Lambda , \tag{136}$$

where $\hat{k} = k_\mu dx^\mu$ and $\hat{v}^\Lambda = v^\Lambda{}_\mu dx\mu$ where $v^\Lambda$ is a spacelike vector normal to $k$. Then

$$k^\mu k^\nu T_{\mu\nu} \overset{\mathcal{H}}{\sim} g_{xy} \iota_k \phi^x \iota_k \phi^y - I_{\Lambda\Sigma} v^\Lambda \cdot v^\Sigma \overset{\mathcal{H}}{\sim} 0 . \tag{137}$$

The second term is non-negative because $I_{\Lambda\Sigma}$ is definite-negative and the vectors $v^\Lambda$ are spacelike in mostly-minus signature. Since $g_{xy}$ is positive-definite, the condition Eq. (56) must be satisfied by each scalar $\phi^x$. Om the other hand, find $v^\Lambda = 0$ and the conditions Eqs. (B.7) must be satisfied for all $\Lambda$, that is

$$\iota_k F^M \overset{\mathcal{H}}{=} 0 . \tag{138}$$

By construction, it satisfies

$$d\mathbf{Q}[\xi] \doteq \Theta(\varphi, \delta_\xi \varphi) + \iota_\xi \mathbf{L}, \tag{144}$$

where in this theory

$$\Theta(\varphi, \delta_\xi \varphi) = -\star(e^a \wedge e^b) \wedge \delta_\xi \omega_{ab} + g_{xy} \star d\phi^x \delta_\xi \phi^y + F_\Lambda \wedge \left(\iota_\xi F^\Lambda + dP_\xi{}^\Lambda\right), \tag{145a}$$

$$\mathbf{L} \doteq -\tfrac{1}{2} F^\Lambda \wedge F_\Lambda - \star V. \tag{145b}$$

For $\xi = k$, where $k$ is some Killing vector

$$\Theta(\varphi, \delta_k \varphi) = -\vartheta_k{}^A \left(K_{Ax} \star d\phi^x - T_A{}^\Lambda{}_M F_\Lambda \wedge A^M\right), \tag{146a}$$

$$\iota_k \mathbf{L} \doteq \tfrac{1}{2} \left(dP_k{}^\Lambda \wedge F_\Lambda + dP_{k\Lambda} \wedge F^\Lambda\right) - \tfrac{1}{2}\vartheta_k{}^A \left(T_A{}^\Lambda{}_M F_\Lambda \wedge A^M + T_{A\Lambda M} F^\Lambda \wedge A^M\right) - \iota_k \star V, \tag{146b}$$

so that

$$\Theta(\varphi, \delta_k \varphi) + \iota_k \mathbf{L} \doteq d\left\{\tfrac{1}{2}\left(P_k{}^\Lambda \wedge F_\Lambda + P_{k\Lambda} \wedge F^\Lambda\right)\right\} + \vartheta_k{}^A J_A - \iota_k \star V, \tag{147}$$

where the on-shell-closed NGZ 3-form currents $J_A$ have the form given in Eq. (122b) for the subset of indices $A, B, \dots$ corresponding to the invariances of the scalar potential.

Substituting this expression in Eq. (144) we find the identity

$$d\left\{\star(e^a \wedge e^b)P_{kab} + \tfrac{1}{2}\left[P_k{}^\Lambda \wedge F_\Lambda - P_{k\Lambda} \wedge F^\Lambda\right]\right\} = \vartheta_k{}^A J_A - \iota_k \star V. \tag{148}$$

As usual, we can define on-shell the 2-forms $\mathcal{J}_A$ and $\mathcal{V}_k$

$$J_A \doteq d\mathcal{J}_A, \tag{149a}$$

$$\iota_k \star V \doteq d\mathcal{V}_k, \tag{149b}$$

whose form depends on the solution on which they are evaluated. Notice that the 2-forms $J_A$ do not satisfy a Gauss law and are different from the 2-form charges $\mathbf{Q}_A[k]$, which depend on the Killing vector and satisfy a Gauss law. The combinations $\vartheta_k{}^A J_A$ for $k = \partial_t, \partial_\varphi$ are just $\omega J_K$ and $m J_L$ and we will call the corresponding 2-forms $\omega \mathcal{J}_K$ and $m \mathcal{J}_L$

The on-shell-closed generalized Komar charges is, therefore,

$$\mathbf{K}[k] = -\frac{1}{16\pi G_N{}^{(4)}} \left\{\star(e^a \wedge e^b)P_{kab} + \tfrac{1}{2}\left[P_k{}^\Lambda F_\Lambda - P_{k\Lambda} F^\Lambda\right] - \vartheta_k{}^A \mathcal{J}_A - \mathcal{V}_k\right\}. \tag{150}$$

We can always derive a Smarr formula by integrating this generalized Komar charge over the relevant boundaries. In boson-star spacetimes, though, it is not clear how to identify generically the results in terms of conserved charges and potentials. Since the electric and magnetic charges vanish, it is likely that the terms containing the gauge field strengths will not contribute to the integrals at infinity, but the other two terms can contribute, explaining the existence of boson star solutions like those constructed in Refs. [48, 49, 56] for instance.

In black-hole spacetimes, since, as we have explained, we must have $K = L$ and $T = S$, we find that

$$\vartheta_k{}^A \mathcal{J}_A = \vartheta_t{}^A \mathcal{J}_A - \Omega_H \vartheta_\varphi{}^A \mathcal{J}_A = (\omega - \Omega_H m)K = 0, \tag{151}$$

by virtue of the synchronization condition Eq. (58) and the next-to-last term in Eq. (150) vanishes identically. As we have discussed, the synchronization condition also ensures that the momentum maps are constant over the horizon and we can proceed as in the symmetric case, obtaining exactly the same Smarr formula Eq. (130). This formula allows for rotating black-hole solutions as those constructed in Refs. [48, 49].

# 6 Discussion

In this paper we have shown how to construct 2-form charges satisfying Gauss laws and how they can be used to restrict (or forbid) the existence of boson-star or black-hole solutions. We have paid special attention to the construction of the generalization of the standard Komar charge of General Relativity from which Smarr formulas can be found. We have considered several cases of increasing complexity leaving outside our scope theories of the Proca–Higgs type which can be constructed by gauging the symmetries of the scalars, eliminating some of them to obtain mass terms trough the Stückleberg mechanism and also Yang-Mills fields. The former give rise to "Proca–Higgs balls", stars and black holes [19, 52] and the later to global monopoles [57–60]. It should be possible to extend the methods developed here to study these two kinds of solutions, as well to extend them to asymptotically-AdS solutions [61] for which one can use the positive energy theorem of Ref. [62]. Work in this direction is already under way [53].

We have also studied the generalized symmetric Ansatz used to construct all the known boson star solutions, considering a generic case with an arbitrary group of isometries and global symmetries. We have argued that this Ansatz should be understood as a different implementation of the spacetime symmetries on the matter fields and not as a breaking of those symmetries ("non-inheritance" [44]).

The obvious similarity that we have found between the integrability condition of this Ansatz and the quadratic constraint of the embedding tensor formalism is quite remarkable and calls for further study. It should be noticed that this Ansatz has been extensively used in the context of generalized dimensional reduction and that it is also related to the construction of "U-folds." These intriguing connections deserve further study since they may lead to a complementary understanding of the reasons for the existence of the boson star and hairy black hole solutions considered in the literature.

# Acknowledgments

TO wishes to thank M.M. Fernández for her permanent support.

**Funding information** This work has been supported in part by the MCI, AEI, FEDER (UE) grants PID2021-125700NB-C21 ("Gravity, Supergravity and Superstrings" (GRASS)), and IFT Centro de Excelencia Severo Ochoa CEX2020-001007-S. The work of RB has also been supported by the National Agency for Research and Development [ANID] Chile, Doctorado Nacional, under grant 2021-21211461 and by PUCV, Beca Pasantía de Investigación.

# A General considerations on the generalized symmetric Ansatz

In this appendix we want to study the generalized symmetric Ansatz for arbitrary isometry and global groups. We will mainly focus on scalar fields, for the sake of simplicity. Thus, we assume

1. The existence of a group of isometries of the spacetime metric generated by the Killing vector fields $k_m \equiv k_m{}^\mu(x)\partial_\mu$, with Lie brackets

$$[k_m, k_n] = f_{mn}{}^p k_p \,, \tag{A.1}$$

which may simply be a subgroup of the complete isometry group. The isometries we are considering may be those of an internal space, if we are interested in a dimensional

compactification Ansatz, or just part of the Ansatz for a boson-star, black-hole or any other kind of solution.

The diffeomorphisms generated by those Killing vector fields act on the matter fields, here represented by scalar fields $\phi^x$ parametrizing some target space, as

$$\delta_m \phi^x = -\pounds_{k_m} \phi^x, \tag{A.2}$$

where $\pounds_{k_m}$ is the Lie derivative with respect to the vector field $k_m$

$$\pounds_{k_m} \phi^x = \iota_{k_m} \phi^x \equiv \iota_m \phi^x. \tag{A.3}$$

2. The existence of a global symmetry group acting on the matter fields. On the scalar fields that we are considering as an example, the generators are

$$\delta_I \phi^x = K_I^x(\phi), \tag{A.4}$$

where the $K_I^x(\phi)$ are Killing vectors of the target space metric $g_{xy}(\phi)$ satisfying the Lie algebra

$$[K_I, K_J] = f_{IJ}{}^K K_K. \tag{A.5}$$

The generalized symmetric Ansatz assumes that the scalar fields are not symmetric under the infinitesimal general coordinate transformations (GCTs) generated by the Killing vectors of the spacetime metric, in the naive sense

$$\pounds_m \phi^x \equiv \pounds_{k_m} \phi^x = 0, \tag{A.6}$$

but in a generalized sense [43]

$$\delta_m \phi^x \equiv -\pounds_m \phi^x + \vartheta_m{}^I K_I{}^x = 0. \tag{A.7}$$

Here the $\vartheta_m{}^I$ are constants (we call them *shift constants*) and the above equation indicates that the scalars are invariant under the GCTs generated by the spacetime Killing vectors up to a global symmetry generated by a certain combination of the target-space Killing vectors $K_I$.

## A.1 Consistency condition

The Ansatz Eq. (A.7) has to satisfy a consistency (or integrability) condition: if we act with the Lie derivative with respect to a different spacetime Killing vector, we find

$$\pounds_m \pounds_n \phi^x = \vartheta_n{}^I \pounds_m K_I{}^x = \vartheta_n{}^I \partial_y K_I{}^x \pounds_m \phi^y = \vartheta_n{}^I \vartheta_m{}^J K_J{}^y \partial_y K_I{}^x, \tag{A.8}$$

and, antisymmetrizing in $m, n$ and using the definition of the Lie bracket, we get, up to a factor of $1/2$,

$$[\pounds_m, \pounds_n] \phi^x = -\vartheta_m{}^I \vartheta_n{}^J [K_I, K_J]^x = -\vartheta_m{}^I \vartheta_n{}^J f_{IJ}{}^K K_K{}^x, \tag{A.9}$$

where we have used Eq. (A.5). Using the fundamental property of the Lie derivative, Eq. (A.1) and the linearity of the Lie derivative, we find that,

$$[\pounds_{k_m}, \pounds_{k_n}] = \pounds_{[k_m, k_n]} = \pounds_{f_{mn}{}^p k_p} = f_{mn}{}^P \pounds_{k_p}, \tag{A.10}$$

and plugging this relation into Eq. (A.9) and using the Ansatz again, we arrive to the following relation between structure constants and shift constants

$$f_{mn}{}^P \vartheta_p{}^K = -\vartheta_m{}^I \vartheta_n{}^J f_{IJ}{}^K. \tag{A.11}$$

We notice that this relation is formally identical to the so-called quadratic constraint of the embedding tensor formalism.[37] In that formalism, the consistency condition Eq. (A.11) can be seen to arise from the requirement that the embedding tensor be invariant. Here, if we view the shift constants $\vartheta_m{}^I$ as objects an adjoint index $m$ and another index $I$ in some other representation r of the spacetime symmetry subgroup, and transforming with matrices

$$\Gamma_{\text{Adj}}(T_m)^p{}_n \equiv f_{mn}{}^p, \qquad \Gamma_{\text{r}}(T_m)^K{}_J \equiv \vartheta_m{}^I f_{IJ}{}^K, \tag{A.12}$$

invariance means

$$\delta_m \vartheta_n{}^I = \Gamma_{\text{Adj}}(T_m)^p{}_n \vartheta_p{}^I + \Gamma_{\text{r}}(T_m)^I{}_J \vartheta_n{}^J = f_{mn}{}^p \vartheta_p{}^I + \vartheta_m{}^J \vartheta_n{}^K f_{JK}{}^I = 0, \tag{A.13}$$

which is the consistency condition Eq. (A.11).

This consistency condition imposes strong constraints on the possible Ansatzs. Let us consider, for example, a 4-dimensional spherically-symmetric spacetime and let us focus on the generators of the SO(3) isometry subgroup with structure constants $f_{mn}{}^p = \varepsilon_{mnp}$, $m, \ldots = 1, 2, 3$. It is evident that the shift constants can only be non-trivial if $f_{IJ}{}^K \neq 0$ and, therefore, one uses for the generalized symmetric Ansatz a non-Abelian subgroup of the target space isometry group. In particular, for the very often considered case of a massive, complex, Klein–Gordon scalar, which only has one available isometry, all the components of the shift constants must vanish identically in the spherically symmetric case and none of the generators of SO(3) can be used to define a generalized symmetric Ansatz. However, if the spacetime is stationary and axisymmetric, those two commuting spacetime symmetries can be combined with the phase shifts of the Klein–Gordon field as in Ref. [49]. This example was studied in Ref. [43].

Observe that a trivial way to satisfy the constraint is to use identical spacetime and target-space Lie algebras and shift constants which are proportional to Kronecker deltas.

## A.2 Scalar charges and the generalized symmetric Ansatz

As we have explained in the main text, in the standard symmetric case, one can construct 2-form scalar charges satisfying a Gauss law using the invariance of all the fields under the diffeomorphisms generated by Killing vector, which implies the invariance of the on-shell-closed Noether–Gaillard–Zumino (NGZ) 3-form currents (see Eq. (42)) $\delta_k J_I$.[38] This leads to the on-shell closedness of the interior products of the Killing vector $k_m$ and the NGZ 3-forms $d\iota_m J_I \doteq 0$, which we can use as 2-form scalar charges $\mathbf{Q}_{mI} \equiv \iota_m J_I$.

This mechanism does not work in the generalized case, in which we have

$$\begin{aligned} \delta_m J_I &= -d\iota_m J_I - \iota_m dJ_I + \vartheta_m{}^J \delta_J J_I \\ &\doteq -d\iota_m J_I + \vartheta_m{}^J f_{JI}{}^K J_K = 0, \end{aligned} \tag{A.14}$$

except when one uses Abelian global symmetry groups.

In the non-Abelian case, though, we can define on-shell 2-forms $B_I$

$$J_I \doteq dB_I, \tag{A.15}$$

that can be seen as the duals of the scalar fields [55,64] and we can rewrite the above expression as a total derivative

$$d\left(\iota_m dB_I - \vartheta_m{}^J f_{JI}{}^K B_K\right) \doteq 0, \tag{A.16}$$

which allows us to define the on-shell-closed 2-form scalar charges

$$\mathbf{Q}_{mI} \equiv \iota_m dB_I - \vartheta_m{}^J f_{JI}{}^K B_K, \qquad d\mathbf{Q}_{mI} \doteq 0. \tag{A.17}$$

---

[37]See, e.g. Eq. (3.57) in Ref. [63].

[38]Here, $J_I$ is the NGZ 3-form current associated to the global generator labeled by $I$.

# B  The (restricted) generalized zeroth law for the electrostatic and magnetostatic potentials

If we define the electrostatic and magnetostatic potentials through the momentum map equations (64) and (66) that we reproduce here for the sake of convenience

$$\iota_k F + d\Phi = 0 \,, \tag{B.1a}$$

$$\iota_k \tilde{F} + d\tilde{\Phi} = 0 \,, \tag{B.1b}$$

using the Killing vector $k = \partial_t - \Omega_H \partial_\varphi$ that satisfies $k^2 \overset{\mathcal{H}}{=} 0$ and $k \overset{\mathcal{BH}}{=} 0$, we immediately find that these potentials are constant on $\mathcal{BH}$:

$$d\Phi \overset{\mathcal{BH}}{=} 0 \,, \tag{B.2a}$$

$$d\tilde{\Phi} \overset{\mathcal{BH}}{=} 0 \,. \tag{B.2b}$$

In Ref. [23] this property has been called the *restricted generalized zeroth law*, since it is a restriction of the standard generalized zeroth law which says that they are constant over the whole $\mathcal{H}$ to $\mathcal{BH}$. The generalized zeroth law follows from the restricted one and from the equations

$$\iota_k d\Phi = 0 \,, \tag{B.3a}$$

$$\iota_k d\tilde{\Phi} = 0 \,, \tag{B.3b}$$

which are obtained by taking the interior product of the momentum maps equations with $k$.

The standard derivation of the generalized zeroth law[39] that does not rely on the existence of a bifurcation surface but needs, instead, the Einstein equations from which Eq. (55) follows. Applying this identity to the energy-momentum tensor of the Maxwell field we find[40]

$$\iota_k F \cdot \iota_k F \overset{\mathcal{H}}{=} 0 \,, \tag{B.4a}$$

$$\iota_k \tilde{F} \cdot \iota_k \tilde{F} \overset{\mathcal{H}}{=} 0 \,, \tag{B.4b}$$

which implies that $\iota_k F$ and $\iota_k \tilde{F}$ are null in $\mathcal{H}$. Since $\iota_k \iota_k F = \iota_k \iota_k \tilde{F} = 0$, the only possibility is that

$$\iota_k F \overset{\mathcal{H}}{\propto} \hat{k} \,, \tag{B.5a}$$

$$\iota_k \tilde{F} \overset{\mathcal{H}}{\propto} \hat{k} \,, \tag{B.5b}$$

where $\hat{k}$ is the 1-form dual to the Killing vector $k$, that is $\hat{k} \equiv k_\mu dx^\mu$. Using the momentum-map equations, it follows that

$$\hat{k} \wedge d\Phi \overset{\mathcal{H}}{=} 0 \,, \tag{B.6a}$$

$$\hat{k} \wedge d\tilde{\Phi} \overset{\mathcal{H}}{=} 0 \,, \tag{B.6b}$$

which, together with Eqs. (B.3) implies that $\Phi$ and $\tilde{\Phi}$ are constant over $\mathcal{H}$.

Here it is important that these functions are defined as momentum maps with respect to $k = \partial_t - \Omega_H \partial_\varphi$. The purely electrostatic and magnetostatic potentials that would have been obtained in $k = \partial_t$ do not have this property. The electric and magnetic charges are the same

---

[39]See, for instance Ref. [65].

[40]The Maxwell energy-momentum tensor is invariant under the replacement of $F$ by $\star F = \tilde{F}$.

on the horizon and at infinity because they satisfy Gauss laws and they no longer need to vanish.

Finally, observe that, as a consequence of the generalized zeroth law,

$$\iota_k F \stackrel{\mathcal{H}}{=} 0\,, \tag{B.7a}$$

$$\iota_k \tilde{F} \stackrel{\mathcal{H}}{=} 0\,. \tag{B.7b}$$

A generalization of these results when there are several gauge and scalar fields can be found in footnote 36 in page 24.

## C  A generalized stationary solution of the Maxwell equations in Minkowski spacetime

In this appendix we want to show how to find solutions of the Maxwell equations and Bianchi identities satisfying the generalized symmetric Ansatz Eqs. (77) with $k = \partial_t$. It is convenient to work with the gauge fields $A, \tilde{A}$ because the Maxwell equations and Bianchi identities are automatically satisfied and we only need to demand the self-duality condition Eq. (75). Thus, we are going to use the formulation of the generalized symmetric Ansatz for gauge fields Eqs. (88). We also make the Ansatz/gauge choice $\chi_k = \tilde{\chi}_k = 0$ that reduces Eqs. (88) to

$$\partial_t \begin{pmatrix} A \\ \tilde{A} \end{pmatrix} = \begin{pmatrix} \omega\tilde{A} \\ -\omega A \end{pmatrix}, \tag{C.1}$$

which, working in Cartesian coordinates, can be solved by

$$A_t = \tilde{A}_t = 0\,, \quad A_m = \cos(\omega t) f_m(x)\,, \quad \tilde{A}_m = \sin(\omega t) f_m(x)\,, \quad m, n, p, \ldots = 1, 2, 3\,. \tag{C.2}$$

The solution will satisfy the self-duality condition Eq. (75) if the time-independent $f_m(x)$ satisfies the equation

$$\varepsilon_{mnp}\partial_n f_p = -\omega f_m \quad \Rightarrow \quad \partial_m f_m = 0\,. \tag{C.3}$$

This equation is solved in spherical coordinates by

$$f_r = 0\,, \qquad f_\theta = \frac{A\cos(\omega r) + B\sin(\omega t)}{\sin\theta}\,, \qquad f_\varphi = -A\sin(\omega r) + B\cos(\omega r)\,, \tag{C.4}$$

where $A$ and $B$ are integration constants.

This solution oscillates in time and space and the components of the gauge fields are sines or cosines of $\omega(t \pm r)$. The components $F_{\theta\varphi}$ and $\tilde{F}_{\theta\varphi}$ vanish identically and so do the electric and magnetic charges. The non-vanishing components of the energy-momentum tensor do not decay fast enough at infinity but they are time-independent, as expected:

$$T_{tt} = T_{rr} \sim -\frac{\omega^2(A^2 + B^2)}{r^2\sin^2\theta}\,, \qquad T_{\varphi\varphi} = -\sin^2\theta\, T_{\theta\theta} \sim 4\omega^2 AB\sin(\omega r)\cos(\omega r)\,. \tag{C.5}$$

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
