# Peer review of "Generalized Komar charges and Smarr formulas for black holes and boson stars"

_SciPost Physics Core, doi:SciPost Phys. Core 8, 038 (2025)_

## Round 1 · Referee Report · Anonymous (Referee 1) · 2024-12-26

Report
The authors use the generalized Komar charge to demonstrate the (non-)existence of solitonic solutions in theories where General Relativity is minimally coupled to bosonic degrees of freedom, as well as to derive Smarr-like relations for black hole solutions in such theories. Over the past decades, the issue of the existence of (non-topological) solitons in different field theories, in both flat and curved spacetime, has been thoroughly discussed. Examples presented by the authors do not appear to contradict or introduce significant novelty compared to results already known in the literature. However, the method outlined in the paper could provide a mechanism to partially automate the process of answering questions related to the existence of solitonic solutions. As noted in the discussion around Eq. 1.22, this promise relies on having an explicit expression for the form ω, which is not necessarily available in general.
Similarly, regarding the derivation of Smarr-like formulas in hairy black hole scenarios, alternative methods have been developed and successfully applied—for instance, the results of Herdeiro et al., cited by the authors. Thus, looking forward, it may be beneficial to understand the comparative advantage of the proposed approach with respect to the methods used so far, both regarding solitonic solutions (e.g., arguments based on Derrick's theorem) as well as the Smarr formula.
Requested changes
1. Usefulness of the method: Regarding the last point of the report, the authors may provide an additional discussion of the advantage of their method compared to the alternatives. For example, on pg. 4, the authors claim that in the case of other methods for deriving the Smarr-like relations in the theories under consideration, "the results are more difficult to understand because they involve unphysical quantities." However, Herdeiro et al., in 1403.2757 and 2406.03552, seem to successfully connect the horizon mass/angular momentum to ADM quantities and use the generalized Smarr formula to validate their numerical results—see the section on "Physical relations and checks" in 1403.2757 and the discussion around Eq. 2.70 in 2406.03552.
2. References: It would be appropriate for the authors to reference the original results regarding the (non-)existence of non-topological solitons in bosonic field theories (both in flat and curved spacetime) and to situate their specific conclusions, as well as the broader aim of the paper, within the context of these results. In particular, I would suggest citing the seminal works of Derrick, Kaup, Coleman, and Lee
https://doi.org/10.1063/1.1704233
https://journals.aps.org/pr/abstract/10.1103/PhysRev.172.1331
https://www.sciencedirect.com/science/article/abs/pii/055032138590286X?via%3Dihub
https://journals.aps.org/prd/abstract/10.1103/PhysRevD.35.3658
3. Terminology: The authors choose to use the loaded term "boson star" instead of the more standard terms "non-topological soliton" or "soliton star" (e.g., doi.org/10.1016/0370-1573(92)90064-7). While harmonizing terminology in the literature is challenging, the authors could consider adding a comment on this matter. Specifically, it is widely assumed that a "boson star" solution already implies a self-gravitating complex scalar or vector field protected by U(1) or some other global
symmetry (e.g. https://arxiv.org/abs/1202.5809). Thus, when the authors claim that "there are no boson star solutions in [real, massless scalar] theory [without the potential]" (pg. 10), they are merely stating a well-known (and trivial) fact: such a theory does not admit non-topological solitons.
Recommendation
Ask for minor revision

---

## Round 1 · Referee Report · Anonymous (Referee 2) · 2025-1-3

Report
The authors set out to investigate the question of existence of solitons in General Relativity minimally coupled to various (bosonic) matter fields. They constrain the space of possible black hole and solitonic solutions by deriving generalized Komar charges and Smarr-like formulae in these theories. The standard Smarr relations applicable to stationary solutions of vacuum General Relativity are modified by additional terms due to the presence of the matter fields and associated charges. The manuscript also considers the case when the stationarity condition for matter fields is implemented in a more general (than the usual) way.
Given that there is significant interest in understanding the space of stationary solutions in gravitational theories, the research question discussed in the manuscript is worthy of consideration. The present work is also is motivated by the fact that an application of Smarr's relation (together with the positive energy theorem) yields nonexistence of gravitational solitons in vacuum General Relativity. However, it appears that in a more general class of theories the generalized Smarr relation is not quite as powerful tool as in vacuum General Relativity. Nevertheless, Smarr-like relations may provide important consistency checks for finding black hole and solitonic solutions in various theories.
The methodology and the presentation of the results is also sound and therefore I can recommend the manuscript for publication, modulo a minor revision.
Requested changes
1. The authors briefly outline some of their notational conventions in the beginning of section 2 (and footnote 18, in particular). However, they already use some non-trivial notation early on in the Introduction which may be a bit confusing for the reader at first. I think it would be helpful to fix notations and conventions already in the Introduction. For example, the vierbein notation is first discussed before eq. (2.2) but it already appears in eq. (1.6). It appears that latin indices are tetrad labels rather than abstract indices, a point which might also be worth clarifying. Furthermore, there are some undefined notations in the manuscript later on: e.g. {\cal D} is not defined in eq. (2.10).
2. The authors may consider studying the review Chrusciel et al., Living Rev.Rel. 15 (2012) 7 in the context of black hole solutions. Section 7.2 may be particularly relevant since additional Smarr-type relations are discussed. The methods presented there could be applicable to a more general class of theories and may be helpful to further constrain the space of possible stationary solutions in those theories.
Recommendation
Ask for minor revision

---

## Round 2 · Referee Report · Anonymous (Referee 2) · 2025-3-5

Report

I am happy with the changes made and I recommend the manuscript for publication.

Recommendation

Publish (meets expectations and criteria for this Journal)

---

## Round 2 · Referee Report · Anonymous (Referee 1) · 2025-3-22

Report

Changes to the draft of the paper make it now adequate for publication.

Recommendation

Publish (meets expectations and criteria for this Journal)

---

## Round 2 · Author Response

Dear editor,

After careful reading of the two referee's reports we have made a number of
changes in the manuscript that we list in our answers to the referees.

Answers to the questions posed by referee 1:

\begin{enumerate}
\item As we have commented in the paper, other methods have been used to
derive Smarr formulae and one may say that they are, in the end, equally
effective, since formulae which are identically satisfied for the relevant
black hole solutions are derived. The same basic relation can be expressed
in terms of different variables )as in the references mentioned by the
referee), but not all variables have the same physical standing.

It is well known that in General Relativity only the total (ADM) mass and
angular momentum, defined by surface integrals at infinity, are conserved
and that there is no invariant definition of local mass/energy or angular
momentum density that allows the assignment of some amount of energy or
angular momentum to a given spacetime region, like the event horizon. It
seems reasonable to expect that meaningful physical expressions should be
exclusively and finally written in terms of them.

The method we use here (which was pioneered by Bardeen, Carter and Hawking
in the new references [5,6], see also the new references [7,8]) only
involves surface integrals and, therefore, establishes a clear separation
between quantities which are defined asymptotically and quantities which are
defined on the event horizon. The former are total total, ADM conserved
charges, some of them (electric and magnetic charges) multiplied by their
chemical potentials evaluated at infinity and the latter are the temperature
and entropy, total electric and magnetic charges and their chemical
potentials evaluated over the horizon. The reason why the total electric and
magnetic charges appear in both integrals is that they satisfy Gauss laws.

In contrast to this, the method used in the references mentioned by the
referee and in Townsend's lectures involves volume integrals. The results of
these volume integrals can be, in the end, may be empirically related to
total, ADM, conserved charges but, by construction, they cannot be
identified with energies/masses. When the charges are volume integrals of
conserved currents associated to global symmetries, they cannot be directly
associated to the black hole (they are computed on their exterior) nor to
the spacetime (they are not the total charges). We believe that it is not
natural to express the Smarr formula (which is nothing bu a Gibbs-Duhem-type
thermodynamical relation) in terms of this kind of charges which are not
thermodynamical variables.

The method used in this paper leads to this final expressions
in a more straightforward way.

We have made several changes in the paper to address this important point:
\begin{enumerate}
\item We have added a 2-page discussion of the charges associated to global
symmetries and computed as volume integrals in the context of black-hole
physics at the beginning of the introduction.
\item We have added the references [5-8] in which the method used in this
paper was used.
\item We have rephrased the paragraph below (1.24) to make our point clear.
\end{enumerate}

\item We have added a long footnote (number 13 in page 8) citing the classical
references suggested by the referee and commenting on their results, in
relation with ours.

\item In this work we do cover boson stars in the sense mentioned by the
referee, although we have probably used it in an unconventional or improper
way in a few places. Readers were warned of this fact in the second
paragraph below equation (1.32). Nevertheless, we have added a footnote in
page 12 (formerly page 10) commenting our terminology.

\end{enumerate}

Answers to the questions posed by referee 2:

\begin{enumerate}
\item We have moved the content of footnote~18 to footnote~5, combining it
with the previous content of that note and extending it a bit. $\mathcal{D}$
was implicitly defined in footnote~18 but we have added a line clarifying
its definition.
\item We have mentioned the reference suggested by the referee as well as
Heusler, Phys. Rev. D, 56, 961–973, (1997) in which the alternative method
to derive Smarr formulas is introduced.
\end{enumerate}

We have made some further changes:

\begin{enumerate}
\item We have corrected some misprints.
\item We have cited the original and earlier works in which the Komar charge
and its generalizations were used to derive Smarr fomulae ([5,6,7,8] in the
revised version).
\end{enumerate}

---

## Round 2 · List of Changes

\begin{enumerate}
\item We have added a 2-page discussion of the charges associated to global
symmetries and computed as volume integrals in the context of black-hole
physics at the beginning of the introduction.
\item We have added the references [5-8] in which the method used in this
paper was used.
\item We have rephrased the paragraph below (1.24) to make our point clear.
\item We have added a long footnote (number 13 in page 8) citing the classical
references suggested by the referee and commenting on their results, in
relation with ours.
\item We have added a footnote in page 12 (formerly page 10) commenting our terminology.
\item We have moved the content of footnote~18 to footnote~5, combining it
with the previous content of that note and extending it a bit. $\mathcal{D}$
was implicitly defined in footnote~18 but we have added a line clarifying
its definition.
\item We have mentioned the reference suggested by the referee as well as
Heusler, Phys. Rev. D, 56, 961–973, (1997) in which the alternative method
to derive Smarr formulas is introduced.
\item We have corrected some misprints.
\item We have cited the original and earlier works in which the Komar charge
and its generalizations were used to derive Smarr fomulae ([5,6,7,8] in the
revised version).

\end{enumerate}

---

## Editorial Decision

published